# Prefrontal cortex neurons encode ambient light intensity differentially across regions and layers

Elyashiv Zangen[1], Shira Hadar[1], Christopher Lawrence[1], Mustafa Obeid[1], Hala Rasras[1], Ella Hanzin[1], Ori Aslan[1], Eyal Zur[1], Nadav Schulcz[1], Daniel Cohen-Hatab[1], Yona Samama[1], Sarah Nir[1], Yi Li[1], Irina Dobrotvorskia[1] & Shai Sabbah ⬚[1] ✉

While light can affect emotional and cognitive processes of the medial pre-frontal cortex (mPFC), no light-encoding was hitherto identified in this region. Here, extracellular recordings in awake mice revealed that over half of studied mPFC neurons showed photosensitivity, that was diminished by inhibition of intrinsically photosensitive retinal ganglion cells (ipRGCs), or of the upstream thalamic perihabenular nucleus (PHb). In 15% of mPFC photosensitive neurons, firing rate changed monotonically along light-intensity steps and gradients. These light-intensity-encoding neurons comprised four types, two enhancing and two suppressing their firing rate with increased light intensity. Similar types were identified in the PHb, where they exhibited shorter latency and increased sensitivity. Light suppressed prelimbic activity but boosted infra-limbic activity, mirroring the regions' contrasting roles in fear-conditioning, drug-seeking, and anxiety. We posit that prefrontal photosensitivity represents a substrate of light-susceptible, mPFC-mediated functions, which could be ultimately studied as a therapeutical target in psychiatric and addiction disorders.

The medial prefrontal cortex (mPFC) orchestrates multifaceted cognitive and emotional processes[1–5]. Decision-making, working memory, mood processing, anxiety, and fear, are mPFC-modulated functions found to be affected by light[6]. Examples include light-dependent enhancement of speed and accuracy in detection tasks associated with enhanced mPFC responses[7], and differential effects of lighting colour on improvements in working memory[7–9] that are also differentially associated with mPFC responses[7,8]. Moreover, manipulation of the transmission of light signals to the mPFC alters depression-like behaviours in animals[10], distinct light colours differentially decrease negative mood and increase amygdala-PFC connectivity in humans[11], and bright light leads to enhanced fear extinction together with suppression of fear acquisition, both being accompanied by light-associated modulation of mPFC activity[12]. We have recently further reported

activity corresponding with the intensity of light in several gross regions of the human mPFC[13]. The substrate of such correspondence however, i.e., an ability of mPFC neurons to encode light-intensity, remains unclear.

Although the mPFC does not receive direct retinal input, light sensitivity may be bestowed indirectly, through a pathway recently demarcated on anatomical grounds[10]. In this pathway, intrinsically photosensitive retinal ganglion cells (ipRGCs) innervate the periha-benular nucleus (PHb), a dorsomedial thalamic region projecting to the mPFC[10,14]. ipRGCs, by expressing the photopigment melanopsin, integrate long-lasting autonomous light sensitivity with rod/cone photoreceptors sensitivity[15,16], forming a specialized retinal output channel dedicated to a stable representation of environmental light[17–20]. Here, by combining light-evoked multielectrode-array

[1]Department of Medical Neurobiology, Faculty of Medicine, The Hebrew University of Jerusalem, Jerusalem 9112102, Israel.
✉e-mail: shai.sabbah@mail.huji.ac.il

recordings from the mPFC and PHb of awake mice, with extensive neuronal mapping and chemo- and optogenetic manipulation, we trace light-intensity encoding in mPFC neurons, and propose a pathway that may drive, at least partly, such intensity encoding.

## Results

While using 32-site multielectrode arrays in the mPFC of awake, head-restrained wildtype mice, we stimulated the eyes with diffused light at 7 intensities spanning a 6-log range, covering the operational ranges of rods, cones, and melanopsin[19,21] (Supplementary Fig. 1a, b). At each intensity, the stimulus comprised twenty 10 s pulses interspersed by 10 s darkness epochs. For processing firing rate (FR), the acquired wide-band neural activity (0–40 kHz) was high-passed filtered at 250 Hz. Submerging the probe in red fluorescence dye enabled mapping the probe in slices using coordinates of a standardized mouse brain and identifying the nearest neuron to each recording site (Fig. 1a). The estimated accuracy was 32, 85, and 39 µm along the rostral-caudal, dorsal-ventral, and medial-lateral axes, respectively (see *Methods*). By accounting for the mapping accuracy, a probability could be assigned to the affiliation of each neuron with a specific mPFC subregion and cortical layer. This enabled assigning recordings to neurons, and neurons to mPFC subregions and layers.

In 60 recording sessions, we have identified 1682 neurons in 20 mice. Neurons exhibited light-evoked responses which were either transient, persistent, or both (for an example of a neuron showing light-evoked firing in response to 7 tested intensities, see Supplementary Fig. 1c,d). These were identified by comparing the time-average FR during the 3-sec preceding the stimulus (baseline), to that of the stimulus' first 1 sec (early window), and last 6 sec (late window). Overall, 57% (951) of the 1682 neurons showed a statistically-significant light-evoked change in FR (paired permutation t-test, FR before vs. during stimulus). Out of all recorded neurons, 21.5% (361) showed a transient response, 13.5% (226) a persistent response, and 22% (364) both response types. The remaining 43% (731) did not exhibit a detectable response to light (Fig. 1b). In total, neurons exhibiting a statistically-significant light-evoked transient response, either with or without a persistent component, represented 43% of recorded cells, while those having a persistent response (with or without a transient response) constituted 35% of identified neurons. For comparison, in control recordings of light-evoked firing in the nearby secondary motor cortex (MOs), the incidence of transient and persistent light-responsive neurons per recording session was significantly lower, accounting for 8% and 9% of identified neurons, respectively (here and elsewhere, see the figure caption for statistics; Fig. 1c,d; see Supplementary Fig. 1e for the incidence of light-responsive neurons per animal),

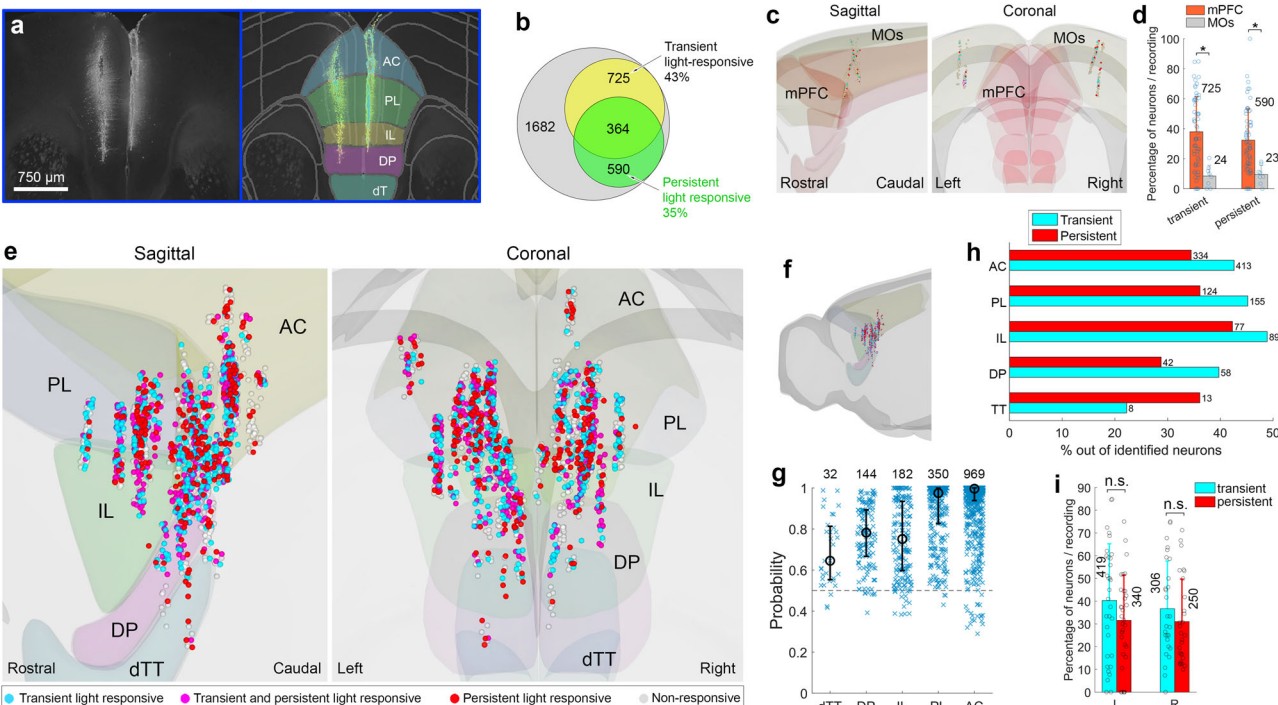

**Fig. 1 | mPFC neurons exhibiting transient and persistent responses to light.**
**a** Two multielectrode array placements (one more medial), during extracellular recordings in awake mice during light stimulation of the eyes, superimposed with corresponding mPFC subregions on a coronal atlas plane. **b** Distribution of neurons exhibiting statistically-significant transient and/or persistent effect of light on FR, or no response. **c** Mapping of neurons identified in the MOs, in sagittal and coronal planes. **d** Incidence (mean ± SD, across recording sessions) of neurons in the mPFC demonstrating transient and persistent light-responsiveness was significantly higher than in the MOs (permutation t-test, one-sided; transient: *p* = 3E-5, effect size (ES): *d* = 1.327; persistent: *p* = 1E-4, ES *d* = 1.145; mPFC: 1682 neurons, 60 recording sessions, 20 mice; MOs: 218 neurons, 8 recording sessions, 2 mice). Here and elsewhere, values over bars represent number of neurons. **e, f** Mapping of neurons in the frontal cortex exhibiting transient and/or persistent light-response (**f**), and a close-up view along the sagittal and coronal planes (**e**). Dots, each representing a neuron, indicate whether the neuron is light-responsive across the 'early' (blue) or 'steady-state' (red) window, or during both (magenta). Non-responsive neurons are coloured in grey. Abbreviations: anterior cingulate, AC; prelimbic cortex, PL;

infralimbic cortex, IL; dorsal peduncular area, DP; dorsal taenia tecta, dTT.
**g** Probability of neurons to be assigned to each mPFC subregion (median, 1st and 3rd quartiles; number of neurons assigned to each subregion are presented).
**h** Distribution (percentage out of recorded neurons) of transient and persistent light-responsive neurons across the mPFC's five subregions. Incidence of transient and persistent neurons in the IL and PL was higher but did not vary significantly compared to the remaining mPFC regions [transient: χ²(1, 1675) = 3.59, *p* = 0.058, one-sided, ES φ = 0. 040; persistent: χ²(1, 1675) = 3.02, *p* = 0.082, one-sided, ES φ = 0.052]. **i** Incidence (mean ± SD) across recording sessions of neurons in the mPFC demonstrating transient and persistent light-responsiveness did not differ significantly across hemispheres [permutation t-test, two-sided; transient, *p* = 0.94; persistent, *p* = 0.99; n_left = 32 and n_right = 28 recording sessions], as did the incidence of neurons across animals [not presented, permutation t-test; transient, *p* = 0.50; persistent, *p* = 0.58; n_left = 18 and n_right = 15 animals]. Mapping of neurons utilized BrainRender https://github.com/brainglobe/brainrender. Source data are provided as a Source Data file.

consistent with previous reports showing MOs innervation by the visual cortex[22-24].

Neurons demonstrating transient and persistent light-responsiveness were identified throughout the mPFC, including the dorsal taenia tecta (dTT), dorsal peduncular area (DP), infralimbic cortex (IL), prelimbic cortex (PL), and anterior cingulate (AC) (Fig. 1e,f), with the median probability for neurons to be affiliated with these subregions being 0.64, 0.78, 0.75, 0.97, and 0.99, respectively (Fig. 1g). These probabilities are roughly correlated with subregion size. The incidence of transient and persistent light-responsive neurons was highest in the PL and IL (transient: 43-49%, persistent: 34-42%, of identified neurons) (Fig. 1h). When only neurons with a probability higher than 0.9 of being affiliated with the respective subregion were included, the incidence of persistent light-responsive neurons in the IL became significantly higher than in the remaining regions (Supplementary Fig. 1f). Details on the selection of appropriate statistical tests can be found in the *Methods'* statistical analysis section (Supplementary Fig. 1l,m). The incidence of light-responsive neurons did not differ significantly across hemispheres (Fig. 1i), but slightly varied in a subregion-specific manner (Supplementary Fig. 1g), and along the mPFC's anterior-posterior axis, being relatively stable and high over ~800 μm spanning 1.3 to 2.1 mm from bregma (Supplementary Fig. 1h). In a different data set, we found no sex differences in the incidence of light-responsive neurons (Supplementary Fig. 1i).

## Persistent light-intensity-dependent response of neurons in the mPFC

While a binary light vs. darkness neural response might induce a binary effect of light on physiology and behaviour, encoding light intensity would allow intensity-dependent, gradual modulation according to light intensity. Thus, we sought to determine whether persistent light-responsive mPFC neurons monotonically change their steady-state FR as a function of light intensity (other non-monotonic intensity-dependent changes in FR might exist, but will not be addressed here). For each neuron, we plotted and fitted to a sigmoid the steady-state FR (time-average FR across the last 6 sec of the stimulus) as a function of light intensity. Neurons whose plotted activity had a root mean square error (RMSE) smaller than the 5th percentile of the null RMSE distribution, were treated as intensity-encoding (Fig. 2a). By this criterion, 253 neurons, representing 15% of identified neurons, demonstrated persistent intensity-encoding (Fig. 2b). By the same criterion, in the MOs, such neurons accounted for 4% of identified neurons (Supplementary Fig. 1j, k).

Principal component analysis with gaussian mixture model clustering, revealed distinct types of neurons having persistent intensity-encoding, with the optimal number of types being 4 (Fig. 2c). In all four types, light-evoked responses comprised an early, transient increase in FR (ON peak), followed by an enduring enhancement, or suppression, of FR (late response component). Two of the neuronal types demonstrated a transient increased FR both when the light went ON and OFF; The ON peak followed in one neuronal type by increased steady-state firing, hereafter 'ON-OFF enhanced' ($E_{on-off}$), and in the other by decreased steady-state firing, hereafter 'ON-OFF suppressed' ($S_{on-off}$). The third type exhibited after the ON peak a slow increase in FR, hereafter 'ON enhanced' ($E_{on}$), and the fourth a decrease in FR, hereafter 'ON suppressed' ($S_{on}$). The baseline-subtracted average steady-state FR of the two 'enhanced' types monotonically increased with light intensity, while FR of the two 'suppressed' types monotonically decreased with increasing light intensity (Fig. 2d). These averaged trends are evident in the FR of individual neurons (Supplementary Fig. 2a–c). The sigmoidal intensity-response fits allowed us to calculate the threshold intensity at a FR criterion of 0.1 Hz, which varied by 0.8 log photons cm$^{-2}$ s$^{-1}$ across the different neuronal types (Fig. 2e).

Upon light onset, the latency of the ON peak was $87.5 \pm 9.6$ ms, while that of the late response ranged in $S_{on-off}$, $E_{on}$, and $S_{on}$ between

0.4 and 2 s (Fig. 2f, g). Latency of $E_{on-off}$, which displayed a prominent ON peak partially overlapping the late response, was not estimated. The light-evoked FR of the two 'enhanced' types increased by 53%–73% from baseline, but decreased from baseline by 37%–38% in the two 'suppressed' types (Fig. 2h). Persistent intensity-encoding neurons were found throughout the mPFC (Fig. 2i), with the PL and IL exhibiting a significantly higher incidence (Fig. 2j; see Supplementary Fig. 2d for neurons with subregion-affiliation probabilities > 0.9, showing similar relative incidence). The incidence of these neurons did not significantly differ between sexes nor between hemispheres, but slight subregion-specific variation across hemispheres was encountered (Fig. 2k, Supplementary Fig. 2e, g).

The distribution of the four intensity-encoding types did not significantly differ between hemispheres (Fig. 2l), but did vary between regions. In the AC, IL and DP, the most prevalent type was $E_{on-off}$; in the PL it was $S_{on-off}$; and $S_{on}$ dominated in the dTT (Supplementary Fig. 2f). The IL was dominated by the two enhancement-response types, while the PL by the two suppression-response types (Fig. 2m). This IL-PL contrast was similar when the incidences of these neurons were calculated out of all identified neurons (Supplementary Fig. 2h), and even further accentuated when only neurons with subregion-affiliation probabilities > 0.9 were inspected (Supplementary Fig. 2i). Surprisingly, we discovered another pair of neighbouring subregions, the DP and dTT, with overall opposite responses to light. Upon light exposure, the FR of most DP neurons is enhanced, while that of most dTT neurons is suppressed (note however the relatively small number of intensity-encoding neurons identified in these two regions) (Fig. 2m). The contrasting prevalence of types between regions led to a contrasting effect of light on the IL and PL, and on the DP and dTT, evident in the averaged light-evoked FR of all neurons identified in a given subregion (Supplementary Fig. 2j). Additionally, light-evoked steady-state firing of intensity-encoding neurons was significantly higher in the IL vs. PL and in the DP vs. dTT (Fig. 2n).

Analysing the darkness spontaneous firing and action potential waveforms suggested that the four types corresponded to mixtures of presumptive pyramidal neurons and interneurons, leaning towards a higher composition of pyramidal neurons (an exception was the $E_{on}$ type that corresponded only with presumptive excitatory neurons) (Supplementary Fig. 3a–k). Presumptive pyramidal neurons and interneurons shared dynamics of light-evoked responses, and encoded the intensity of light, with their 'suppressed' types showing a smaller effect of light on FR, possibly reflecting a 'floor effect' (Supplementary Fig. 3l, m). A lower spontaneous FR of neurons relative to interneurons, and the inability of light-evoked FR to decrease below zero, might have limited the range over which light can modulate the two 'suppressed' neuronal types' FR. A smaller light-evoked effect in neurons is consistent with reports of thalamocortical projections onto inhibitory interneurons and excitatory neurons with heavier innervation of inhibitory interneurons, leading to stronger stimulus-evoked effects on inhibitory interneurons[25-30].

## Light intensity continuously modulates mPFC neuronal activity

Environmental light varies gradually rather than in discrete steps. Therefore, we sought to explore whether the identified types could modulate their FR continuously in response to intensity gradients. We recorded light-evoked firing from the mPFC for 7 light intensities as before, followed by 20 repetitions of a bi-phasic stimulus in which log light intensity linearly increased ('ascending phase') and then decreased ('descending phase'). The phases occupied together 60 sec, spanning a 2-log intensity change. Responses were collected from 431 neurons (13 recording sessions, 6 mice). Of those, 28.5% (123) were found to be persistent light-responsive, approximately half of which (55) encoding the intensity of light according to the sigmoid-fit criterion (applied to the 7-intensity stimulus). These neurons were assigned to the four functional types according to the optimal

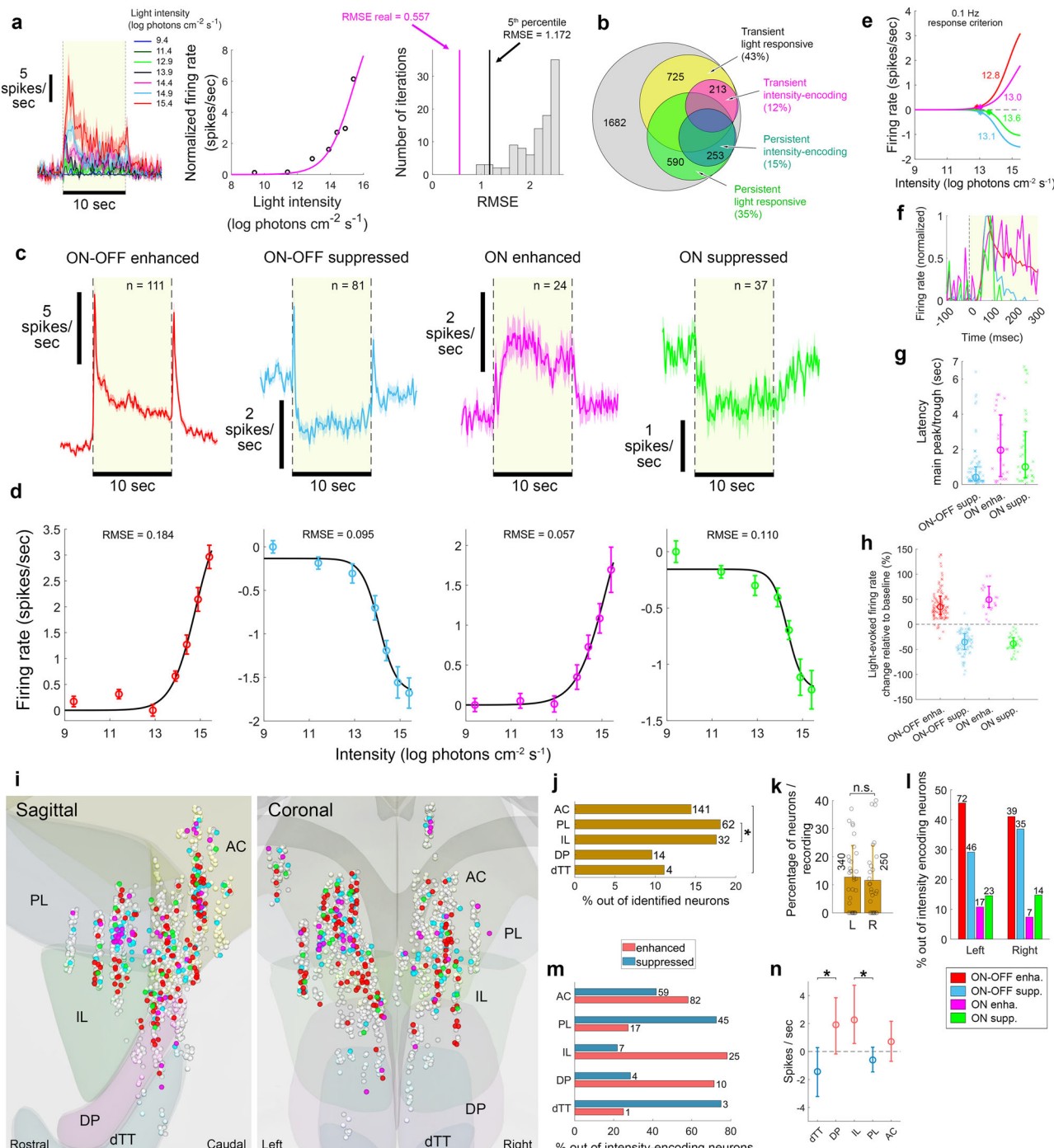

Gaussian mixture model applied above (Fig. 3a). $E_{on-off}$ and $E_{on}$ neurons responded to the bi-phasic stimulus with gradual increase in FR, reaching 39% and 81% increase relative to baseline, followed by gradual decrease; $S_{on-off}$ and $S_{on}$ neurons responded with an inverse response, reaching 14% and 41% decrease relative to baseline (Fig. 3b). Responses to the bi-phasic stimulus showed no transient increases in FR, which tracked smoothly the changing intensity for all types. Exceptions to this trend were local minima/maxima in FR observed during either the ascending or descending phases in all types (Fig. 3b, c), possibly reflecting the switching between photopigment subsystems and/or light adaptation within each subsystem[31,32].

To evaluate each type's capacity to track intensity gradients, we fitted a sigmoid to the FR encountered throughout each phase, for each type (Fig. 3c). The fit was statistically significant for all four types (following comparison between the real sigmoid's RMSE and the null RMSE distribution; see *methods*), demonstrating they all continuously encode light intensity (Fig. 3c). Mean FR transition lagged behind the ascending-to-descending transition by 1 and 0 sec for the two ON-OFF types, and by 3 and 5 sec for the two ON types (Fig. 3c), suggesting that the contribution of prior light exposure to the instantaneous FR of $E_{on}$ and $S_{on}$ is greater than for $E_{on-off}$ and $S_{on-off}$. Indeed, plotting the FR vs. light intensity for each type, revealed that FR during the ascending phase was lower by up to 1 log unit than during the descending phase (Fig. 3c, insets), and that FR after the transition time (30-37 sec from trial onset) was significantly higher than prior the transition time (23–30 s from trial onset) (Supplementary Fig. 4a–d). This supports an effect of light history on mPFC neuronal intensity-encoding. The capacity to continuously encode light intensity was also evident in the FR of individual neurons (Supplementary Fig. 4e–h).

**Fig. 2 | Four types of light-intensity-dependent persistent response in the mPFC. a** Mean ± SEM FR vs. time for 7 intensities (left); steady-state FR fitted to a sigmoid and the RMSE (middle); RMSE null distribution (right). Neurons with $RMSE_{real} < 5^{th}$ percentile $RMSE_{null}$ were deemed intensity-encoding. **b** Distribution of neurons exhibiting statistically-significant transient and/or persistent intensity-encoding response (60 recording sessions, 20 mice). **c** Light-evoked FR (mean ± SEM) during the highest tested intensity, for four functional types. **d** Mean ± SEM baseline-subtracted steady-state FR, vs. light intensity, for the four types. **e** Threshold intensity (asterisk) at FR criterion of 0.1 Hz, varied by 0.8 log photons $cm^{-2} s^{-1}$ across types (colours as in (**c**)). **f** ON peak latency. **g** Type-specific latency of late response (sample as in **c**). Latency differed across types [permutation ANOVA; F = 15.7, $p < 0.001$, effect size (ES) $\eta^2 = 0.094$; median ($25^{th}$ and $75^{th}$ percentile): 'ON-OFF-suppressed': 0.4 (0.2, 1.0) sec, 'ON-enhanced': 1.95 (0.45, 3.95) sec, 'ON-suppressed': 1.0 (0.37, 3.0) sec]. Post-hoc comparisons, permutation t-test, two-sided: latency differed between 'ON-OFF suppressed' and the two 'ON' types ($p = 3E-4$, ES $d = 0.609$), but not across 'ON' types ($p = 0.3206$, ES $d = 0.258$). **h** Percent change in FR from baseline (median, $25^{th}$ and $75^{th}$ percentile) for all types (sample as in (**c**)). Error calculation included all data; data points exceeding ±150% change not plotted, to facilitate comparisons. **i**, Mapping across the mPFC. **j** Persistent intensity-encoding neurons incidence, was higher in the PL and IL [$\chi^2(1, 1675) = 4.36$, $p = 0.036$, one-sided, ES $\varphi = 0.051$]. **k** Incidence (mean ± SD) across recording sessions did not differ between hemispheres [premutation t-test, two-sided; $p = 0.714$, ES $d = 0.095$; $n_{left} = 32$, $n_{right} = 28$ recording sessions], as did the incidence across animals [not presented, premutation t-test, $p = 0.162$, ES $d = 0.503$; $n_{left} = 18$ and $n_{right} = 15$ mice]. **l** Distribution of the four types did not differ between hemispheres [$\chi^2(9, 252) = 2.14$, $p = 0.543$, one-sided, ES Cramer's V = 0.092]. **m** 'Enhanced' vs. 'suppressed' types distribution across mPFC subregions [IL vs. PL: $\chi^2(1, 93) = 22.90$, $p = 2E-6$, one-sided, ES $\varphi = 0.497$; DP vs. dTT: $\chi^2(1, 18) = 2.82$, $p = 0.09$, one-sided, ES $\varphi = 0.396$]. **n** Median ($25^{th}$ and $75^{th}$ percentile) light-evoked steady-state firing of intensity-encoding neurons, across mPFC subregions [sample as in (**j**)]. Light-evoked firing differed across subregions (permutation ANOVA; F = 9.28, $p = 4E-6$, ES $\eta^2 = 0.131$), and was higher in IL vs. PL (permutation t-test, one-sided, $p = 1E-6$, ES $d = 1.152$) and in DP vs. dTT (permutation t-test, one-sided, $p = 0.0079$, ES $d = 1.477$). Mapping of neurons utilized BrainRender https://github.com/brainglobe/brainrender. Source data are provided as a Source Data file.

## Tracking the source of mPFC light-sensitivity

Since the cortical layer of a neuron would suggest anatomical and functional attributes, we calculated for each neuron a probability of being located in a specific cortical layer. The probability of neurons' correct assignment to layer 1 was often below 0.5, hence no interpretation of layer 1 data was attempted. In contrast, probability of neurons to be correctly assigned to all other layers was high, with at least 75% of neurons having a probability higher than 0.5 to be affiliated correctly (Supplementary Fig. 5a). Most identified neurons were located in layer 5 (Supplementary Fig. 5b). The incidence of persistent light-responsive and intensity-encoding neurons was highest in layer 6 of the AC, in layers 5/6 of the PL, and in layer 2/3 and layer 6 of the IL (Supplementary Fig. 5c, d). In the AC, neurons in layers 2/3 were predominantly suppressed by light, while those in layers 5/6 were mostly enhanced by it (Supplementary Fig. 5e). In contrast, across all layers of the IL, 'enhanced' neurons predominated. Since layer 2/3 contains neurons receiving thalamo-cortical innervation, and layer 6 harbours cortico-thalamic feedback neurons[33], the incidences noted above may indicate that persistent light-responsive and intensity-encoding neurons of the AC and PL are predominantly cortico-thalamic feedback neurons, while IL ones represent thalamo-cortical projection neurons as well.

## Dependence of mPFC photosensitivity on input from the presynaptic PHb

The PHb innervates the IL, and to a lesser extent the PL[10], implicating the PHb as a source of light-intensity signalling to the mPFC. To test whether the PHb contributes to mPFC photosensitivity, we chemogenetically inhibited mPFC-projecting PHb neurons, by bilateral injection in the mPFC of a retrograde Cre/GFP-expressing adeno associated virus (AAV), and injection in the PHb of a Cre-dependent AAV-mCherry/designer receptors exclusively activated by designer drugs (DREADDs). We then recorded light-evoked activity from the mPFC following subcutaneous infusion of clozapine N-oxide (CNO) or saline (Fig. 4a). To control for CNO effects on mPFC photosensitivity, we recorded mPFC light-evoked activity following CNO or saline infusion in mice that do not express DERADD in the PHb. Figure 4b, c show examples of mCherry/DREADDs-positive somata of mPFC-projecting PHb neurons, and GFP/Cre-positive somata and axons in the mPFC, sparing the striatum, yet another target of the PHb (for examples of GFP/Cre-positive somata and axons in the mPFC, the orbitofrontal cortex, and the insular cortex, with mCherry/DREADDs-positive axons only in the mPFC, and coronal sections showing mPFC-projecting PHb neurons see Supplementary Fig. 6a–h). While only 2 of 11 mice (2 recording sessions) yielded meaningful data, i.e., high accuracy in all four injections, an optimal balance between AAV incubation time and DREADD expression, and stable recordings, an effect on mPFC photosensitivity following inhibition of mPFC-projecting PHb neurons was observed. Incidence of persistent light-responsive mPFC neurons following CNO infusion was significantly lower than following saline infusion in the DREADDs group (12 vs. 3 neurons); the incidence of such neurons did not significantly differ between saline and CNO infusion in the control group (56 vs. 42 neurons) (Fig. 4d). Thus, these data present a preliminary indication for PHb-mediated contribution to light-responsiveness of the mPFC.

To further test whether PHb input contributes to mPFC photosensitivity, we employed an optogenetic approach. To inhibit the axon terminals of mPFC-projecting PHb neurons, we virally expressed in the PHb an inhibiting, light-activated, G protein-coupled receptor, the eOPN3[34]. By inserting an optotrode into the mPFC, we photoactivated terminals of axons originating in eOPN3-expressing PHb neurons to achieve their continuous inhibition, and simultaneously recorded light-evoked firing in mPFC neurons (Fig. 4e) (see Methods). As a control, we used the same optotrode and photoactivation/recording protocol in mice in which the fluorescent reporter mScarlet, instead of eOPN3, was virally expressed in PHb neurons. Figure 4f and Supplementary Fig. 7a show selective expression of eOPN3 in the PHb, and labelled axon terminals in the IL/PL (where PHb axons terminate[10]). Since the pathway in question is hypothesized to transmit light-intensity signals, light contamination from opto-inhibition may interfere with its activity (see *Methods*). We therefore did not compare light-evoked mPFC FR before vs. during opto-inhibition, but recorded light-evoked mPFC FR under sustained eOPN3-induced inhibition. Absolute steady-state FR of all identified neurons in eOPN3-expressing mice was significantly lower than in mScarlet-expressing mice for the 3 highest intensities (Fig. 4g). Consistent with PHb-dependent mPFC photosensitivity, the incidence of persistent light-responsive and intensity-encoding neurons per recording session in the eOPN3 group was significantly lower than in the mScarlet group (Fig. 4h), as the incidence of neurons belonging to each of the four mPFC intensity-encoding neuronal types (Fig. 4i).

Together, chemogenetic and optogenetic inhibition suggest a contribution of PHb input to mPFC photosensitivity. However, the surviving light-responsiveness and intensity-encoding (Fig. 4, Supplementary Fig. 7b), which may reflect incomplete chemo- and optogenetic inhibition, may also demonstrate the contribution of other sources to the observed mPFC photosensitivity. Moreover, since the white stimulus light might have penetrated the brain and activated the eOPN3 expressed in the PHb and its targets (see Methods), our opto-inhibition cannot exclude an indirect PHb-to-mPFC drive through, e.g., the NAc and dorsomedial striatum, both innervated by the PHb[10].

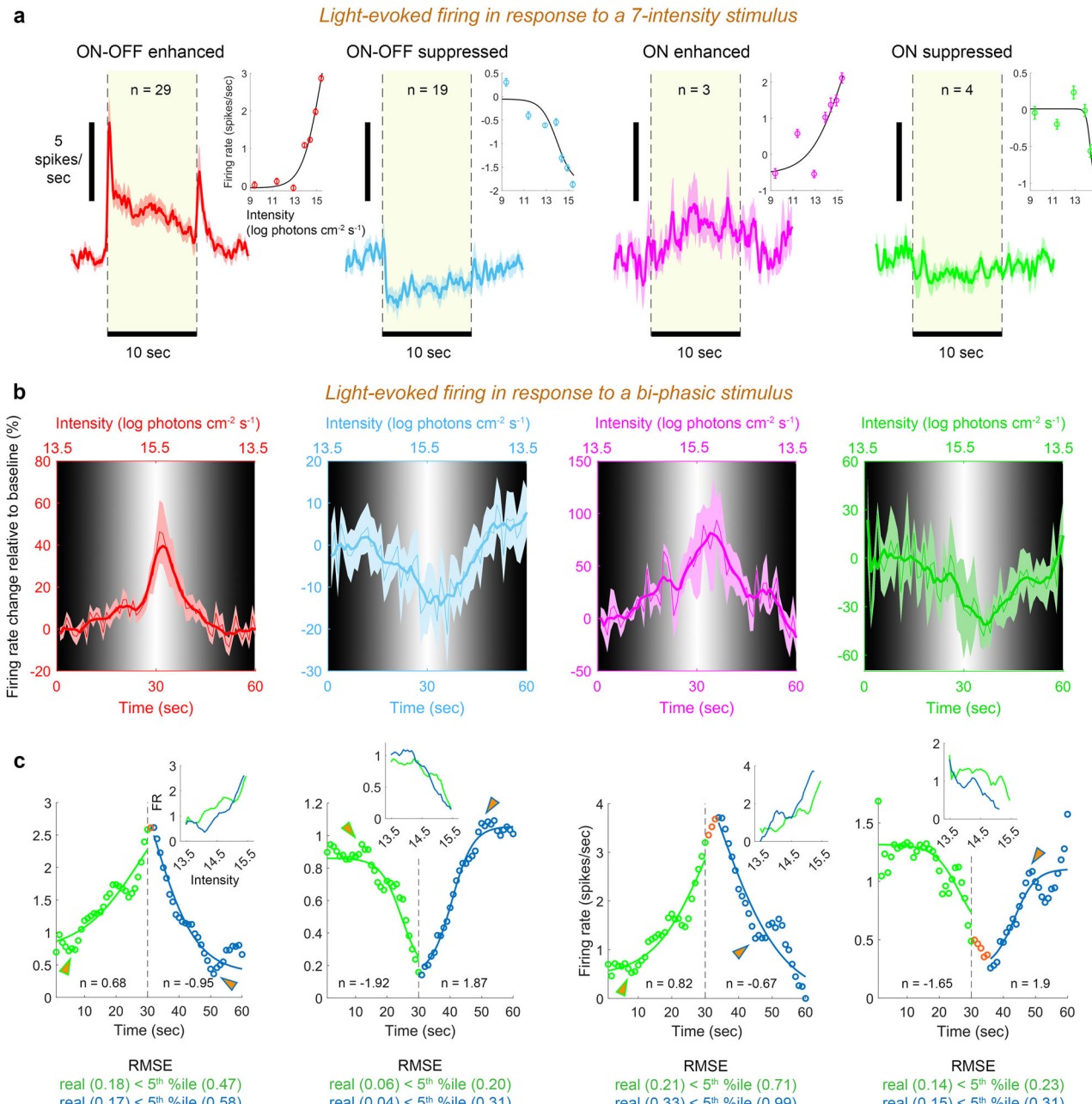

**Fig. 3 | The activity of mPFC neurons continuously tracks the intensity of ambient light. a** Main plots, Light-evoked FR (mean ± SEM across neurons) in response to 10 s of the highest tested intensity, for the four functional types of intensity-encoding neurons. Insets, Steady-state FR (mean ± SEM across neurons) vs. light intensity for the four types. **b** FR modulations (mean ± SEM across neurons) in the four types in response to a bi-phasic stimulus. The secondary (top) abscissa and the grayscale gradient represent light intensity. A smoothed version (thick line; moving average, 5 s) of the raw, noisy, FR trace (thin line) is presented. **c** A sigmoid fitted to the mean FR along each phase, for each type. The transition point from ascending to descending intensity (30 s) is marked by a vertical dashed black line; slopes (n) of the sigmoid fitted to the two phases are indicated; $RMSE_{real}$ and the 5th

percentile of the $RMSE_{null}$ are indicated below plots. Orange: points in which FR change lagged behind the transition between light phases. Considering that each data point represents a 1 s interval, FR lagged behind the intensity transition by 1, 0, 3, and 5 s, for ON-OFF enhanced, ON-OFF suppressed, ON enhanced, and ON suppressed types, respectively. Local minima/maxima in FR (arrows) for all four types appeared along the ascending phase at intensity 13.8 – 14.2 log photons cm$^{-2}$ s$^{-1}$ (6–12 s from stimulus onset), and along the descending phase at intensities 14–14.3 log photons cm$^{-2}$ s$^{-1}$ (46–51 s from stimulus onset). **Inset**. Mean FR (spikes/ sec) as a function of light intensity (log photons cm$^{-2}$ s$^{-1}$) for the ascending and descending phases, for each of the four types. Source data are provided as a Source Data file.

## Light-evoked activity in the presynaptic PHb resembles that in the mPFC

We next inspected light-evoked activity in the PHb, and tested whether it resembles that in the mPFC (Fig. 5a). Since the PHb is not included in any available mouse brain atlas, precise mapping of neurons was not attempted. Instead, to determine the volume occupied by mPFC-projecting PHb neurons (without inhibiting those neurons), we utilized

in a subset of mice the dual-viral approach employed for chemogenetic inhibition. Figure 5b shows an example electrode track spanning that volume; recorded neurons were not necessarily those projecting to the mPFC.

We identified 410 neurons in the PHb (16 recording sessions, 9 mice). Of those, 43% and 31% respectively showed a statistically-significant transient and persistent effect of light on FR, and, 12%

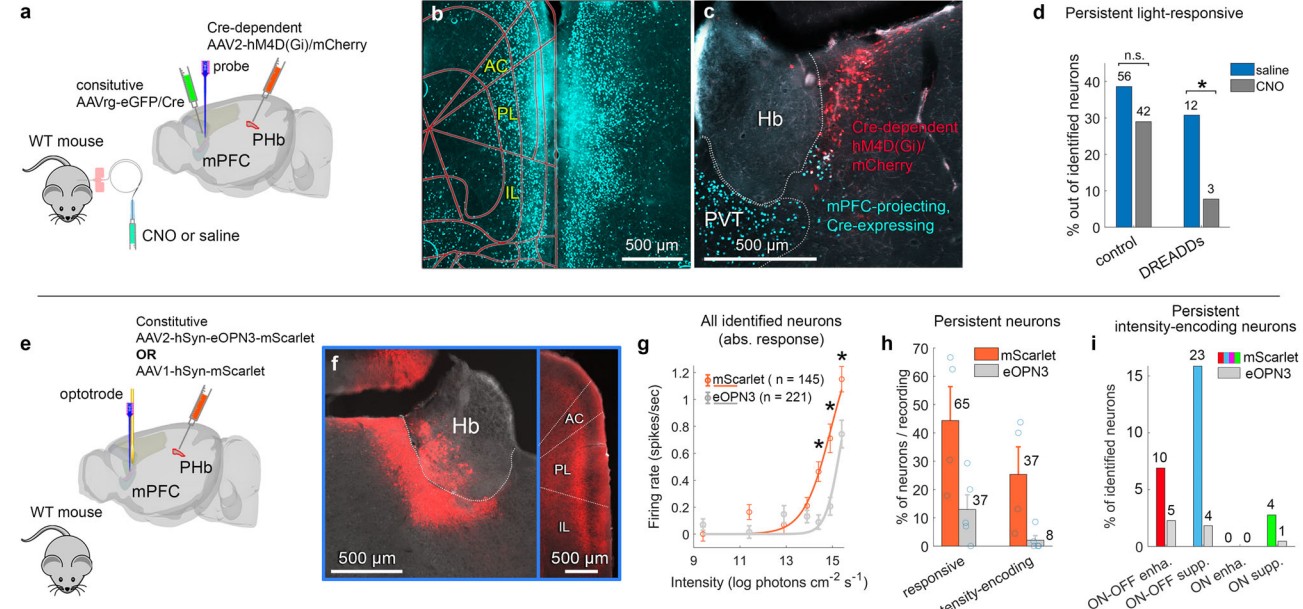

**Fig. 4 | Transmission from PHb affects light-evoked mPFC firing rate. a** Viral labelling of mPFC-projecting PHb neurons in WT mice, by bilateral injection of retrograde Cre/GFP-expressing AAV in the mPFC (shown in the AC, PL), and Cre-dependent AAV-hM4D(Gi) DREADD/mCherry in the PHb. Light-evoked firing of the same neurons was recorded following saline and CNO subcutaneous infusion. **b**, **c** GFP/Cre-positive somata and axons in the mPFC (**b**) and mCherry/DREADDs-positive somata of mPFC-projecting PHb neurons (**c**); reproduced in 4 mice in which all four injections were accurate. **d** Incidence of light-responsive mPFC neurons following saline and CNO, in mice expressing DREADDs in the PHb, and in mice that do not (control). Incidence differed between saline and CNO in the DREADDs group [$\chi^2(1, 78) = 6.68$, $p = 0.009$, one-sided, effect size (ES) $\varphi = 0.102$] but not in controls [$\chi^2(1, 290) = 3.02$, $p = 0.08$, ES $\varphi = 0.293$]. Only 3 intensity-encoding neurons were identified after saline, none retained intensity-encoding following CNO. **e** WT mice bilaterally injected in the PHb with an inhibiting constitutive light-activated eOPN3, to optogenetically inhibit axon terminals of mPFC-projecting PHb neurons using an optotrode, while recording mPFC neuronal firing. Control: mice injected with a constitutive fluorescent reporter. **f** eOPN3/mScarlet

florescence in the PHb, with slight expression in the dentate gyrus (molecular layer) and habenula (both not reported to project to IL/PL). Labeled terminals dominated in the IL/PL (targets of mPFC-projecting PHb neurons). **g** Absolute steady-state FR (mean ± SEM) of identified neurons, vs. light intensity, with fitted sigmoid, in mScarlet and eOPN3 mice (mScarlet, $n = 145$ neurons, 4 sessions, 2 mice; eOPN3, $n = 221$ neurons, 5 sessions, 3 mice). FR in response to the 3 highest intensities in eOPN3 mice was lower than in mScarlet mice (permutation t-test, one-sided, corrected for multiple comparisons; $p = 0.028$, $0.005$, and $0.006$, ES $d = 0.251$, $0.401$, and $0.381$, for the highest to 3rd highest intensities). **h** Incidence across recording sessions (mean ± SD) of light-responsive and intensity-encoding neurons in the eOPN3 group ($n = 5$ sessions) was lower than in controls (mScarlet; $n = 4$ sessions; permutation t-test, one-sided; $p = 0.012$, ES $d = 0.794$ for light-responsive neurons, $p = 0.008$, ES $d = 1.216$ for intensity-encoding neurons), as did the incidence across animals (not presented, permutation t-test, one-sided; $p = 0.016$, ES $d = 4.468$ for light-responsive neurons, $p = 9E-6$, ES $d = 4.567$ for intensity-encoding neurons). **i** Incidence of the four mPFC intensity-encoding types in eOPN3 and mScarlet mice. Source data are provided as a Source Data file.

transiently and 13% persistently encoded light intensity, by the criterion applied to the mPFC (Fig. 5c). Light-evoked FR, when averaged across all identified neurons, recapitulated the bulk light-evoked calcium signal of PHb neurons previously recorded using fiberphotometry[14] (Fig. 5d). Principal component analysis and gaussian mixture model clustering, yielded the optimal number of 4 types (Fig. 5e), resembling those identified in the mPFC, and denoted similarly, $E_{on-off}$, $S_{on-off}$, $E_{on}$, and $S_{on}$. Steady-state FR vs. light intensity sigmoid-fitting, revealed that the four PHb types encode light intensity (Fig. 5f). However, they differed from their mPFC counterparts in several parameters: *i*. The amplitude of light-evoked FR was higher in the PHb, for all types (Fig. 5e, f). *ii*. The latency of the ON peak was 40 ms shorter than in the mPFC (Fig. 5e, insets). *iii*. Compared to the mPFC, light-evoked responses persisted longer after the light stimulus (Fig. 5e) and decay time (from light offset until FR returned to baseline) was longer for all types (Fig. 5g). *iv*. The sensitivity of the four was 1.24–2.73 log photons cm$^{-2}$ s$^{-1}$ higher than of their mPFC counterparts (Fig. 5h). *v*. the baseline FR of all types was significantly higher than for the mPFC counterparts (Fig. 5i). *vi*. the percent change in light-evoked FR relative to baseline was smaller than in the mPFC (Fig. 5j). The latter two findings explain the observation that the baseline-subtracted light-evoked firing in the PHb is larger than in the mPFC (Fig. 5e, f). *vii*. Finally, the relative prevalence of the PHb types differed significantly from that of mPFC types (Fig. 5k): while $E_{on-off}$ neurons were the most

abundant in both regions, $S_{on-off}$ neurons represented the second largest fraction in the mPFC but the smallest fraction in the PHb; instead, $S_{on}$ neurons represented the second largest fraction in the PHb.

## ipRGCs underlie mPFC photosensitivity

Transmission of ipRGC signals through the PHb, or other brain regions, might underlie the intensity-encoding of the mPFC. To test this, we chemogenetically inhibited ipRGCs in the Opn4$^{Cre/+}$ mouse, and recorded neuronal activity in the mPFC in response to 7 light intensities after subcutaneous infusion of CNO or saline (7 DREADD-expressing mice, 18 recording sessions, 312 light-responsive neurons). In control experiments, we expressed in ipRGCs only mCherry, and recorded neuronal activity following CNO or saline infusion (3 mCherry-expressing mice, 6 recording sessions, 56 light-responsive neurons). Figure 6a,b show the experimental protocol and an example retina with DREADD/mCherry-positive ipRGCs.

Among light-responsive neurons, we compared the FR change in response to the highest light intensity in DREADD-expressing vs. mCherry-expressing neurons, following saline vs. CNO infusion. This revealed a significant effect of CNO in DREADD-expressing mice, but not in mCherry-expressing mice (see figure caption for statistical analysis) (Fig. 6c). Among intensity-encoding neurons, we compared the FR in response to the 7 tested light intensities, in DREADD-expressing vs. mCherry-expressing mice, after saline vs. CNO infusion.

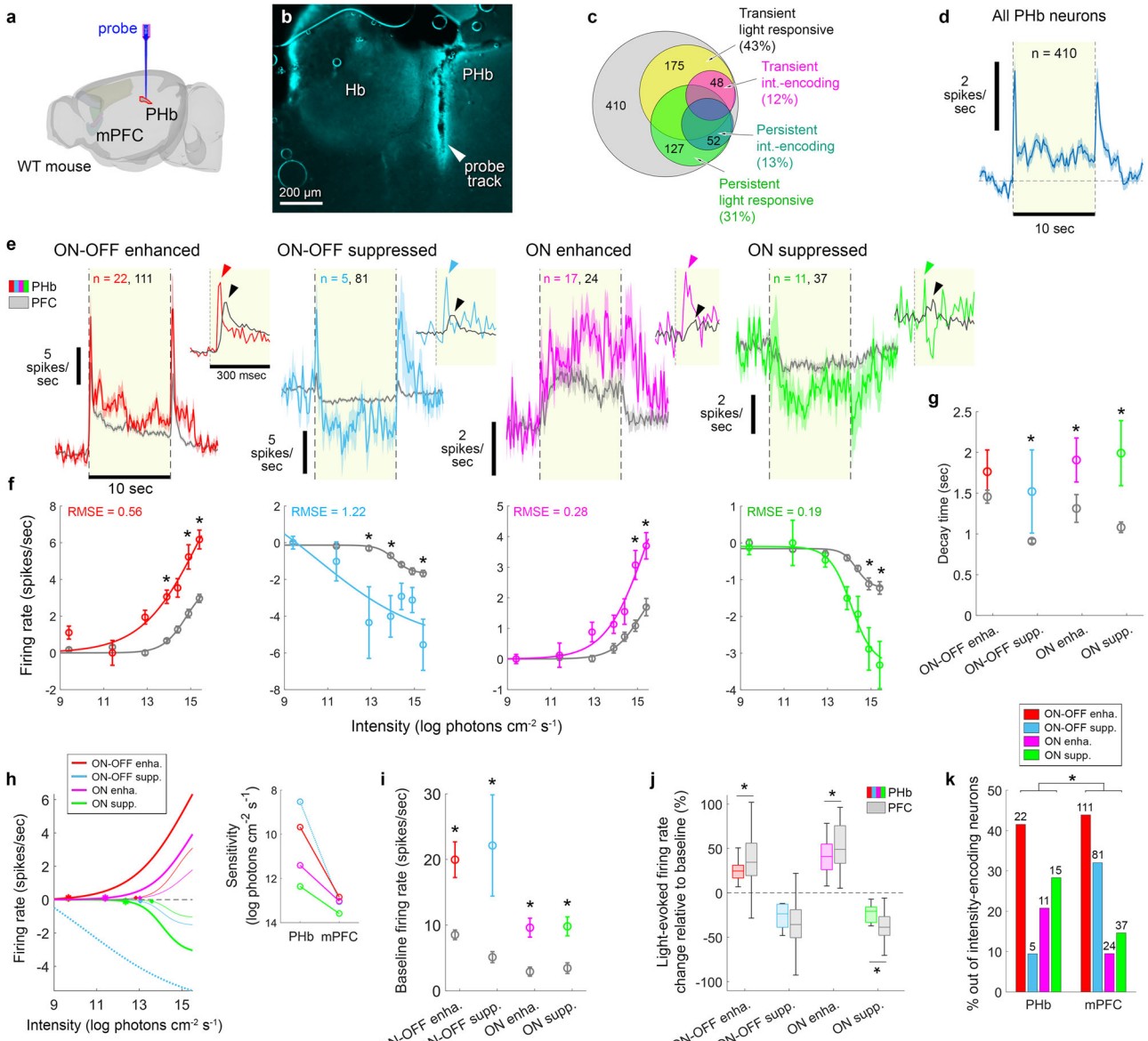

**Fig. 5 | The PHb harbours light-intensity-encoding neurons. a**, **b** Schematic (**a**) and image (**b**) of an extracellular probe in the PHb. **c**, Incidence of PHb transient and persistent types, among light-responsive and intensity-encoding neurons. **d** Mean ± SEM FR across responsive and unresponsive neurons in the PHb. **e** Light-evoked FR (mean ± SEM) for the PHb's four types (colour) and their PFC counterparts (grey). Insets, FR at 10 ms instead of 100 ms binning; the ON peak in the PHb (47.5 ± 5 ms) arrives 40 ms earlier than in the mPFC (87.5 ± 9.6 ms), for all types except 'ON-suppressed'. **f** For all types, amplitude (mean ± SEM) of light-evoked FR in the PHb vs. mPFC differed for a subset of the 7 tested intensities (permutation t-test, two-sided; asterisks: $p < 0.05$, corrected for multiple comparisons; 'ON-OFF-enhanced', $p = 0.0076$, 0.0246, and 0.001 for the highest, and $2^{nd}$ and $4^{th}$ highest intensities; 'ON-OFF-suppressed', $p = 0.0054$, 0.0048, and 0.0006 for the highest, and $4^{th}$ and $5^{th}$ highest intensities; 'ON-enhanced', $p = 0.0180$ and 0.008 for the highest and $2^{nd}$ highest intensities; 'ON-suppressed', $p = 0.0114$ and 0.0372 for the highest and $2^{nd}$ highest intensities). **g** Decay time (mean ± SEM) of light-evoked firing in the PHb vs. mPFC was longer for all types, and statistically significant for the 'ON-OFF-suppressed', 'ON-enhanced', and 'ON-suppressed' (permutation t-test, one-sided; asterisks: $p = 0.0084$, 0.0302, and 0.0011; effect size (ES) Cohen's $d = 0.88$, 0.63 and 1.06; sample as in (**e**)). **h** Light sensitivity for all types in the PHb

(thick lines) and mPFC (thin lines). A 0.1 Hz response threshold criterion was attained at 9.67, 11.4, and 12.36 log photons cm$^{-2}$ s$^{-1}$ in the PHb, compared to 12.4, 13.02, 13.05, and 13.6 log photons cm$^{-2}$ s$^{-1}$ in the mPFC. For 'ON-OFF-suppressed', quality of fit was low (high RMSE); thus no estimation of response threshold was attempted. Inset, PHb sensitivity was 2.73-, 1.65-, and 1.24 log photons cm$^{-2}$ s$^{-1}$ higher than in the mPFC, for 'ON-OFF-enhanced', 'ON-enhanced', and 'ON-suppressed', respectively. **i**, Baseline FR (mean ± SEM) in the PHb was higher than in the mPFC for all types; asterisks: $p < 0.05$; (permutation t-test, one-sided; $p = 0.0001$, 0.0011, 0.0002, and 0.0004; ES Cohen's $d = 1.12$, 1.37, 1.63, and 1.21; for 'ON-OFF-enhanced', 'ON-OFF-suppressed', 'ON-enhanced' and 'ON-suppressed', sample as in (**e**)). **j** Light-evoked change in FR relative to baseline (median; box: $25^{th}$, $75^{th}$ percentile; error bars: $10^{th}$, $90^{th}$ percentile) was smaller in the PHb than in the mPFC for all types except for 'ON-OFF-suppressed' (permutation t-test, one tail, $p = 0.027$, 0.125, 0.043, and 0.0019; ES Cohen's $d = 0.505$, 0.617, 0.661 and 1.164 for 'ON-OFF-enhanced', 'ON-OFF-suppressed', 'ON-enhanced', and 'ON-suppressed'; asterisks: $p < 0.05$). **k** Distribution of PHb types differed significantly from that of their mPFC counterparts [$\chi^2(9, 306) = 17.69$, $p = 5E-4$, one-sided, ES Cramer's V = 0.241]. Source data are provided as a Source Data file.

In DREADD-expressing mice (36 neurons), absolute FR increased monotonically with light intensity following saline infusion, but less so following CNO infusion; the difference in FR between saline and CNO reached statistical significance in the 4 highest intensities. In mCherry-

expressing mice as well (31 neurons), FR differed significantly between CNO and saline for the 3 highest intensities, yet still increased roughly at a similar rate with light intensity for either saline or CNO infusion (Fig. 6d). Therefore, the observed decrease in intensity-encoding

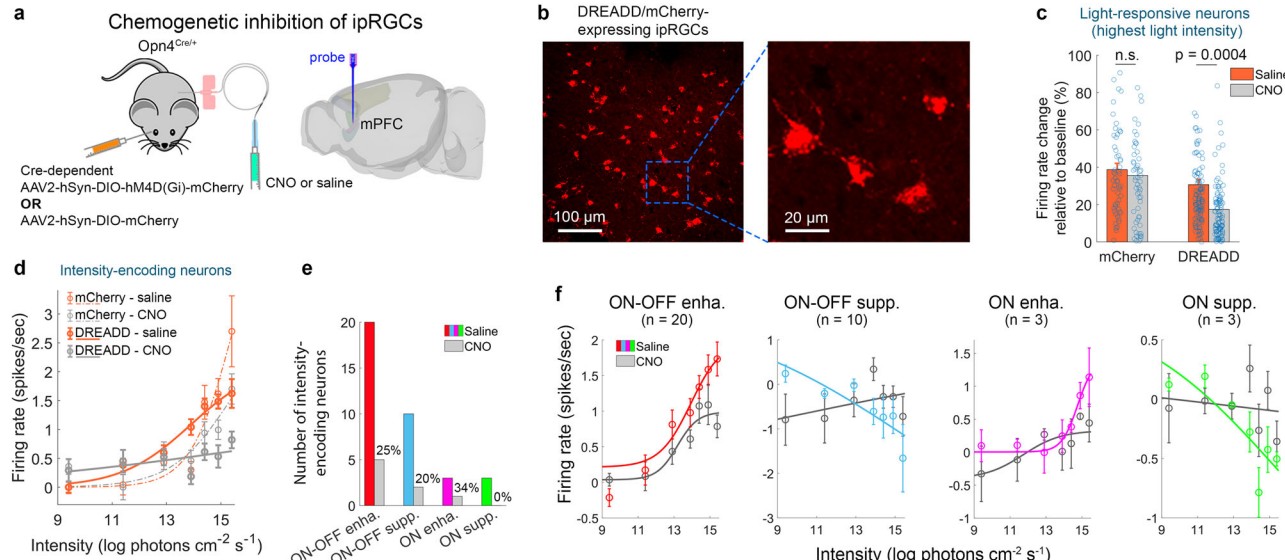

**Fig. 6 | Chemogenetic inhibition of ipRGCs disrupts mPFC light responsiveness and intensity-encoding. a** Chemogenetic inhibition of ipRGCs in the Opn4$^{Cre/+}$ mouse. **b** Example retina with DREADD/mCherry-positive ipRGCs. **c** Absolute FR change relative to baseline (mean ± SEM; in response to the highest intensity), of light-responsive neurons ($n = 101$ and 56) in DREADD-expressing vs. mCherry-expressing mice, following infusion of saline vs. CNO. Error calculation included all data; data points exceeding 100% change not plotted, to facilitate comparisons. Effect of CNO vs. saline on the relative change in FR was evident in DREADD-expressing, but not in mCherry-expressing mice (permutation ANOVA; F = 6.2, $p = 4E-4$, effect size (ES) $\eta^2 = 0.094$). DREADD-CNO differed from all other groups (permutation t-test, two-sided; DREADD-CNO vs. DREADD-saline, $p = 0.002$, ES $d = 0.555$; DREADD-CNO vs. control-CNO, $p = 1E-4$, ES $d = 0.713$; DREADD-CNO vs. control-saline, $p = 1E-4$, ES $d = 0.953$). DREADD-saline, control-CNO, and control-saline groups did not differ between one another (permutation t-test, two-sided; DREADD-saline vs. control-CNO, $p = 0.837$, ES $d = 0.164$; DREADD-saline vs. control-saline, $p = 0.819$, ES $d = 0.300$; control-CNO vs. control-saline, $p = 0.545$, ES $d = 0.111$). **d** Absolute steady-state FR (mean ± SEM) of intensity-encoding neurons (all four types pooled) as a function of light intensity, along with the fitted sigmoid,

in DREADD-expressing ($n = 36$ neurons) vs. mCherry-expressing ($n = 31$ neurons) mice, following saline vs. CNO infusion. FR in DREADD-expressing mice in response to the 4 highest intensities differed significantly between CNO and saline infusion (permutation t-test, two-sided, corrected for multiple comparisons; $p = 0.004$, 0.002, 0.001, and 0.002; ES $d = 0.556$, 0.859, 0.797, and 0.911; for the highest to 4$^{th}$ highest intensity). FR in mCherry-expressing mice in response to the 3 highest intensities differed significantly between CNO and saline infusion (permutation t-test, corrected for multiple comparisons; $p = 0.045$, 0.002, and 0.003; ES $d = 0.391$, 0.533, 0.412; for the highest to 3$^{rd}$ highest intensity). Thus, ES for the highest 3 intensities was higher in DREADD-expressing than mCherry-expressing mice. **e** Number of neurons assigned to each type, following CNO, as compared to saline. Incidence of neurons retaining intensity-encoding firing following CNO is presented above each type's bar ($n = 20$, 10, 3 and 3 neurons). **f** Steady-state FR (mean ± SEM) of the four types of persistent, intensity-encoding neurons, per light intensity (9.4−15.4 log photons cm$^{-2}$ s$^{-1}$) and sigmoid fit, following CNO (grey) vs. saline (coloured). Numbers of neurons is indicated above plots. Source data are provided as a Source Data file.

following CNO is due to chemogenetic inhibition of DREADD-expressing ipRGCs as well as a DREADD-independent effect of CNO.

Clustering by the optimal gaussian mixture model following chemogenetic inhibition of ipRGCs via CNO, showed that ipRGCs inhibition reduced mPFC light-encoding, such that only between 0% and 34% of mPFC neurons belonging to each type retained their persistent light-intensity-encoding firing (Fig. 6e). Fitting steady-state FR vs. light intensity to a sigmoid, revealed that CNO abolished the dependency of FR on light intensity for mPFC neurons of the types $S_{on-off}$, $E_{on}$, and $S_{on}$ (Fig. 6f). In the case of $E_{on-off}$, dependency of responses on light intensity was retained following CNO, although FR decreased (Fig. 6f, left panel). The CNO effect on ipRGC activity was characterized using whole-cell current-clamp recordings from DREADD/mCherry-positive morphologically-identified ipRGCs in flat-mounted retinas (Supplementary Fig. 8). CNO effect varied across cells, but in all tested ipRGCs, light-evoked response decreased, and the dynamic range over which the cells encoded light intensity, narrowed.

To further assess these findings, we terminally ablated ipRGCs. Opn4$^{Cre/+}$ mice were bilaterally intravitreally injected with an AAV inducing Cre-dependent expression of diphtheria toxin A fragment ('dtA'), and constitutive mCherry (Fig. 7a, b), and compared to (1) Opn4$^{Cre/+}$ mice injected bilaterally with an AAV inducing Cre-dependent expression of mCherry ('mCherry'), (2) naïve wild type mice (main data set, 'WT'). In transfected retinal regions of dtA mice, melanopsin-immunopositive ipRGCs were scarce (Supplementary Fig. 9a, b). Extracellular recording of light-evoked responses identified

473 neurons in the 'dtA' group (6 mice, 15 recording sessions), and 158 neurons in the 'mCherry' group (3 mice, 6 recording sessions).

The incidence of mPFC light-responsive neurons and intensity-encoding neurons in the 'dtA' group was significantly lower than in the 'WT' and 'mCherry' groups, compared to no significant difference between the 'WT' and 'mCherry' groups (statistics in figure caption) (Fig. 7c). Therefore, dtA-induced terminal ablation of ipRGCs significantly reduces the incidence of mPFC light-responsive and intensity-encoding neurons, demonstrating the contribution of ipRGC drive to mPFC photosensitivity. Among all identified neurons, we compared FR in response to the 7 light intensities in the three groups. The intensity-dependence of FR in the WT and mCherry groups was virtually identical and greater than in the dtA group, suggesting an effect of dtA ablation on the overall capacity for mPFC intensity encoding (Fig. 7d). dtA-induced ablation of ipRGCs affected the four mPFC intensity-encoding types differentially. It decreased the incidence of $E_{on-off}$ and $S_{on-off}$ by 2-3-fold, and the incidence of $E_{on}$ and $S_{on}$ by 5-10-fold (Fig. 7e). Thus, most of the intensity-encoding neurons that survived the dtA-induced ablation belonged to either $E_{on-off}$ or $S_{on-off}$, but almost non to the more persistent $E_{on}$ and $S_{on}$ types (Supplementary Fig. 9c shows the FR of the few neurons that survived dtA ablation, compared to neurons identified in the wildtype mouse; main data set). Together, our chemogenetic inhibition and terminal ablation experiments suggest that the persistent intensity-encoding capacity of mPFC neurons of the types $S_{on-off}$, $E_{on}$ and $S_{on}$ arises predominantly if not solely from ipRGCs photosensitivity. The retention of intensity-

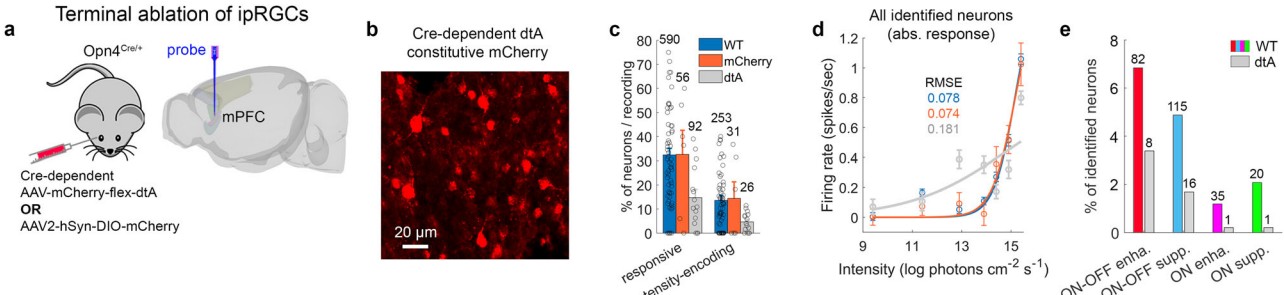

**Fig. 7 | Terminal ablation of ipRGCs diminishes mPFC light responsiveness and intensity-encoding. a** Terminal ablation of ipRGCs in the Opn4$^{Cre/+}$ mouse. **b** Example retina following injection of an AAV inducing Cre-dependent expression of dtA and constitutive expression of mCherry. **c** Incidence (mean ± SEM) of light-responsive and intensity-encoding neurons per recording session. The dtA (15 recording sessions), mCherry (6), and WT (60) groups differed in incidence of light-responsive neurons but not of intensity-encoding neurons (permutation ANOVA, F = 4.71 and 2.29, p = 0.012 and 0.108, ES $\eta^2$ = 0.110 and 0.072, respectively). Pairwise comparisons: incidence across sessions of light-responsive neurons and intensity-encoding neurons in dtA mice was lower than in WT and mCherry mice (permutation t-test, one-sided, light-responsive: p = 0.002 and 0.034, ES d = 0.895 and 1.020; intensity-encoding: p = 0.034 and 0.032, ES d = 0.699 and 1.005), and

WT and mCherry mice did not differ between one another (light-responsive: p = 0.982, ES d = 0.068, intensity-encoding: p = 0.8950, ES d = 0.181). The three groups did not differ in the incidence across mice of light-responsive or intensity-encoding neurons (permutation ANOVA, p > 0.07; number of mice: 6, 3, and 20 in the dtA, mCherry, and WT groups). **d** Absolute steady-state FR (mean ± SEM) of all identified neurons as a function of light intensity, along with the fitted sigmoid, in the WT, mCherry, and dtA groups (n = 1681, 158, and 473 neurons). FR in response to the 2 highest intensities in the WT group was significantly higher than in the dtA group (permutation t-test, one-sided, corrected for multiple comparisons; p = 0.001 and 0.006; ES d = 0.218 and 0.150; for the highest and 2nd highest intensities). **e** Incidence of the four intensity-encoding types, in the dtA vs. WT groups. Source data are provided as a Source Data file.

encoding by E$_{on-off}$ neurons suggests a possible contribution from conventional RGCs. This differential impact on different functional mPFC neuronal types, may suggest a connectivity distinction between the types.

## Discussion

### Origin of intensity-encoding in the mPFC

The FR of 15% of neurons identified in the mPFC persistently encodes discrete light-intensity steps, and closely tracks light intensity gradients. These neurons, mostly corresponding to presumptive excitatory neurons, are divisible into four functional types, two enhancing and two suppressing their FR with increasing light intensity. This mPFC light-responsiveness and intensity-encoding depend almost exclusively on ipRGCs' photosensitivity. However, since ipRGCs lack an OFF response in their firing[15,35,36], the OFF response of the two mPFC ON-OFF types might reflect drive from conventional OFF or ON-OFF RGCs. mPFC photosensitivity also depends on signals transmitted from the PHb, which harbours four functional types mirroring the mPFC types, yet exhibiting shorter latency, longer decay, and higher sensitivity and response amplitude. However, as discussed below, the dependency of the mPFC's photosensitivity on ipRGCs signalling and PHb drive, does not preclude contributions from other brain areas and other RGC types.

In both the PHb and the mPFC, light-evoked firing comprises an early, transient increase (ON peak), followed by a longer enhancement, or suppression. The latency and sensitivity of the two components suggest that the ON peak is driven by rod/cone photosensitivity, and the late response is also driven by melanopsin photosensitivity[19,37]. Additionally, while the PHb neuronal types encode a wide range of intensities, mPFC neuronal types respond to light and encode its intensity only at high intensities, that correspond to the sensitivity ranges of cones and melanopsin[19]. Therefore, it may be hypothesized that low-intensity light signals are filtered between the PHb and the mPFC, rendering the mPFC insensitive to low-intensity light fluctuations. Moreover, all PHb types exhibit the ON peak, possibly reflecting a directly-transmitted excitatory ipRGC drive, whereas only the two 'suppressed' types display a late decrease, potentially reflecting ipRGC drive indirectly-transmitted through PHb inhibitory interneurons. Previous retrograde transsynaptic tracing showed that ipRGCs feed the PHb-mPFC pathway[10], and retrograde tracing showed that the PHb is innervated by the M1 and M4 ipRGC types[10,38]. However, the

contribution of specific ipRGC types to mPFC photosensitivity is currently unknown.

Considering that excitatory neurons represent ~74% of PHb neurons[38], and that extracellular recordings tend to detect more neurons than interneurons (neurons typically having larger and more easily detectable spikes[39]), the neuronal populations we detected in the PHb are likely dominated by excitatory neurons. This, together with the similarity between the PHb and mPFC types, and previous indications that mostly excitatory PHb neurons innervate the mPFC[10,38], suggests that responses of mPFC types are likely driven by the responses of their PHb counterparts. In a manner consistent with such connectivity, the PL and IL, previously reported to be innervated by the PHb[10], displayed the highest incidence of persistent light-responsive and intensity-encoding neurons. In contrast, the incidence of such neurons in layer 2/3 of the AC, DP, and dTT, which receive little if any PHb input, was low, consistent with non-PHb light signalling obtained through intercortical connectivity or from other brain regions.

In addition to the PHb, ipRGC signalling might be transmitted to the mPFC through two other areas – the central amygdala (CeA) and the lateral hypothalamus (LH)[40-43]. The AC is also innervated by the visual cortex[22], through which it may attain its photosensitivity. However, the AC shares its visual cortex innervation with the secondary motor cortex (MOs)[22-24], which we found to display limited photosensitivity and intensity-encoding capabilities, questioning the significance of the contribution of visual cortex input to the AC's intensity-encoding capacity. Based on ours and previous results[10,22,24,38,40,42-50], we propose a tentative working model for the neural network underlying mPFC photosensitivity (Fig. 8).

The responses of the mPFC to sensory stimuli are, by definition, secondary in nature, as the mPFC does not receive any direct sensory input. Instead, the mPFC integrates input from a variety of brain regions[23], with the activity in some of them, e.g., the CeA and LH (Fig. 8), being modulated by light exposure[47,51,52]. Additionally, light alters a multitude of physiological processes and behaviours, which in turn may modulate mPFC activity. For example, light may increase fear by modulating activity in the mPFC-projecting CeA[47,53,54], which in turn may affect mPFC activity[42].

### PL/IL contrasted photosensitivity

Light exposure led to IL excitation along with PL inhibition. The inverse of this duality is reminiscent of the 'PL-go/IL-stop' hypothesis, holding

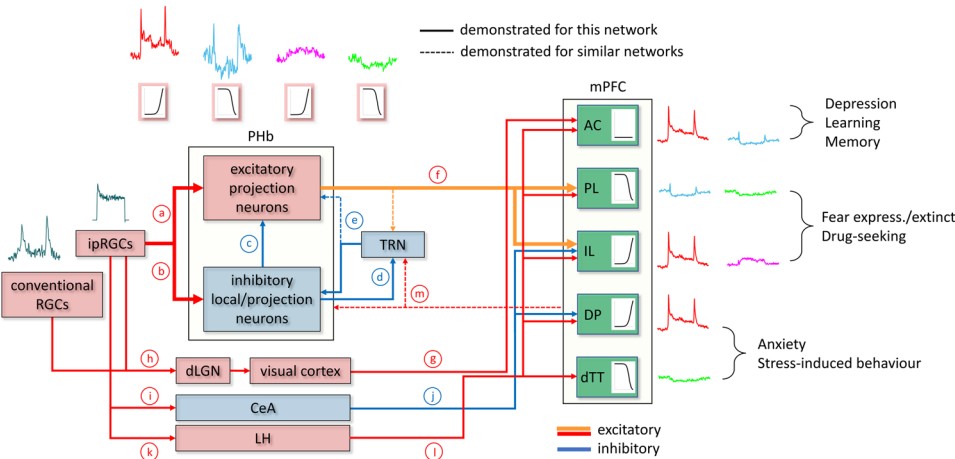

**Fig. 8 | The postulated neural network underlying mPFC photosensitivity.**
**a, b** In the mouse, ipRGCs innervate PHb excitatory relay neurons and inhibitory neurons, which comprise both local interneurons and long-range projecting inhibitory neurons. **c,** Within the PHb, local interneurons synapse on excitatory relay neurons. **d, e** PHb projecting inhibitory neurons innervate the thalamic reticular nucleus (TRN) (**d**), which in turn, may provide inhibitory feedback to some or all of the PHb neuronal types (**e**), as it does to other dorsothalamic areas. **f** PHb excitatory relay neurons innervate the mPFC's IL, PL, and possibly the rostral AC (cg1), perhaps with axon collaterals to the TRN, as often exhibited by neurons in other medial thalamic nuclei. **g, h** The AC receives additional retinal input indirectly through the visual cortex (**g**), which blends input from both ipRGCs and conventional RGCs, transmitted through the dorsal division of the lateral geniculate nucleus (dLGN) (**h**). **i–l** The IL and DP receive additional inhibitory input from the ipRGC-recipient central amygdala (CeA) (**l, j**), and all five mPFC subregions appear to receive additional excitatory input from the ipRGC-recipient lateral hypothalamus (LH)

(**k, l**). **m** Corticothalamic feedback from the mPFC to the PHb, including axon collaterals to the TRN, may also exist. Selected light-modulated behaviours, and the different mPFC regions and intensity-encoding neuronal types that may underlie these behaviours, are indicated. The mFPC harbours four distinct functional types of intensity-encoding neurons, two types enhance and another two types suppress their FR with increasing light intensity. Roughly 80% of the neurons belonging to each type are excitatory neurons, and the rest are inhibitory interneurons (the $E_{on}$ type deviates from this trend as it includes only excitatory neurons). The mPFC includes two pairs of neighbouring subregions that overall, oppositely react to light exposure – light overall enhances IL firing but suppresses PL firing, and similarly, light overall enhances DP firing but suppresses dTT firing. The AC displays equal fractions of 'enhanced' and 'suppressed' types. The PHb harbours four functional types resembling their mPFC counterparts in general form, but exhibit shorter latency, larger amplitude, higher sensitivity, and longer decay time.

that the PL is needed for execution/expression of behaviours, whereas the IL is necessary for the inhibition/extinction of these behaviours[55,56]. In fear expression for example, the PL retrieves the associative memory and expresses the conditioned fear response, while the IL facilitates the retention of fear memory extinction through extinction-induced neuroplasticity[55,57]. Similarly, in animal models of drug addiction, PL activation drives drug seeking while IL activation inhibits it[55,58,59]. Thus, our results raise the possibility that light stimuli may have the effect of inhibiting fear conditioning and drug-seeking (PL inhibition), while promoting the extinction of these behaviours (IL excitation). Indeed, human neuroimaging showed that bright light exposure reduced fear conditioning and increased fear extinction, in tandem with increased mPFC activation during conditioning but with decreased activation during extinction[12].

### Light effects on the dorsal peduncular (DP), the dorsal taenia tecta (dTT), and the anterior cingulate cortex (AC)

Contrasting effects of light were also seen in the net activity of the DP and dTT, through inverse incidence of 'enhanced' and 'suppressed' intensity-encoding types in these regions. Originally considered parts of the olfactory peduncle, the DP and dTT are now recognized as the ventral mPFC, shown to modulate stress-induced and anxiety-like behaviours[60,61], while activation of the DP alone reduces these behaviors[62]. Based on our findings, light exposure can be hypothesized to cause simultaneous enhancement of net DP activity and the suppression of net dTT activity, and may consequently reduce anxiety-like and stress-induced behaviours.

In the current study, the AC harboured the largest number of intensity-encoding neurons, with the incidence of 'enhanced' and 'suppressed' neurons in L2/3 vs. L5/6 being inverted, i.e., intensity-encoding neurons in L2/3 were predominantly suppressed by light, while neurons in L5/6 were mostly enhanced by it. L2/3 contains

intratelencephalic neurons that project to other cortical areas, L5 includes pyramidal tract-like neurons that target subcortical regions, and L6 contains corticothalamic neurons projecting to the thalamus. Hence, light exposure might inhibit the AC's cortical projections through L2/3 projections, but excite the AC's subcortical and thalamic targets through L5/6 projections.

### Correspondence between light- intensity-encoding in the mouse mPFC with human PFC

The light-intensity encoding in the mouse mPFC is consistent with our recent neuroimaging report of light-intensity-dependent alteration in blood-oxygen level-dependent (BOLD) signals in five regions of the human PFC[13]. While a BOLD signal has lower spatial and temporal resolution, the current findings articulate a set of testable hypotheses on regions relationships and connectivity that may be examined in studies on the human PFC. The homology between the rodent and human mPFC is debatable[63], yet the rodent PL and IL are generally thought to correspond with the human pregenual AC and the subgenual AC, respectively[63,64]. Accordingly, light-induced suppression of net activation in the mouse PL would be consistent with the activity suppression we reported in the equivalent human pregenual AC. Light-induced enhancement of net activation in the mouse IL however, is inconsistent with the lack of activity modulation reported in the equivalent human subgenual AC.

The ipRGCs drive to the mPFC we report in the mouse, raises the question of whether intensity-encoding in the human PFC is also driven by ipRGC signalling. A preliminary indication for such transmission is obtained from the finding that the time course of sustained light-evoked human PFC responses, and their susceptibility to prior light exposure, resemble those of ipRGCs[13]. As ipRGCs are especially sensitive to blue light through their melanopsin photopigment[15,16], further testing of this possibility may be achieved through examining the

effect of different-coloured light on PFC activation (indeed, blue light was found to modulate more efficiently BOLD activation in the PFC than other colors[65,66]). While ipRGC signalling might be transmitted to the mPFC through the visual cortex, CeA, LH[22,40–43], and PHb, the latter has never been reported in humans. If a human PHb-analog exists, a probable location would be bounded by the dentate gyrus, lateral habenula, and the central-lateral nucleus of the thalamus, where it is positioned in mice[10,14]. BOLD 7-Tesla functional MRI, previously applied for detection of small brain nuclei[67,68], might allow exploring the existence of a PHb analogue in humans. It likewise remains to be clarified to what extent light-evoked responses in nocturnal species as the mouse recapitulate those encountered in the diurnal human.

As the above-mentioned discussion demonstrates, translation from animal research to humans is non-trivial[69,70]. Nonetheless, a functional link between light exposure and PFC-mediated affective phenomena may be conjectured, based on prior observations in mice implicating a PHb-to-mPFC/NAc transmission in shaping the influence of light on depression-like behaviours[10,14], and the results reported here that expose light-driven activity in the same affective frontal areas, complementing our low-resolution findings in humans[13]. An intriguing avenue would be to harness the obtained knowledge of the characteristics of mPFC light responsiveness, to study human light-modulated PFC processing of emotion and cognition. Specifically, our demonstration that light suppresses PL activity but enhances IL activity, mirrors these mPFC regions' contrasting roles in fear-conditioning, drug-seeking, and anxiety, underlining a potential capacity of carefully-designed light therapy protocols for alleviating anxiety and addiction disorders. Moreover, uncovering the neural mechanisms underlying mPFC light responsiveness might be utilized for the enhancement of cognitive performance through the design of optimal lighting environments.

## Methods
### Animals
Male and female adult (2–4 months old; 23–30 g) WT mice (C57BL/6 J, Jackson Laboratory) and Opn4[Cre/+] mice expressing Cre recombinase in ipRGCs[35] (a generous gift from David Berson), were housed under a reversed 12 h/12 h light/dark cycle with the lights turned off at 8:00 AM, to allow all surgeries and recordings to be conducted during the mice waking hours. Mice were housed at a temperature of 22 °C and 30–60% humidity in groups of 3–4, with food and water ad libitum and available nesting/enrichment material. All experimental procedures were approved by the Authority for Biological and Biomedical Models at the Hebrew University. While we used females and males interchangeably, unfortunately, the sex of mice comprising the main dataset (of mPFC neuronal firing in response to 7 intensity steps) was not documented. Nonetheless, to look into potential sex differences in mPFC photosensitivity, we performed an additional, limited, set of experiments while documenting the sex of the mice (Supplementary Figs. 1i, 2e). Additionally, the particular phase of the oestrous cycle, which lasts four days in mice, was not documented. However, since each mouse underwent 2–4 recording sessions spanning a week, recordings were likely conducted at multiple phases of the oestrous cycle.

### Survival surgery, craniotomy and headplate attachment
Mice were anesthetized with isoflurane (3% in room air; SomnoSuite), injected subcutaneously with an analgesic (buprenorphine, 0.2 mg/kg), and secured in a robotic stereotaxic apparatus (Neurostar). Body temperature was regulated with a heating pad, and depth of anaesthesia was monitored by testing the hind-paw withdrawal reflex and observing respiration. Isoflurane concentration was gradually adjusted downward toward 1.5% over the course of the surgery to maintain a steady plane of anaesthesia. Eyes were covered with sterile ophthalmic ointment (chloramphenicol, 5%) to protect the cornea. The scalp was

shaved, sterilized with iodine, and treated with the local anaesthetic lidocaine. The scalp was then incised and the connective tissue cleared from the calvarium. Bregma and lambda were identified, and the target coordinates were computationally corrected for variation in size and orientation of the mouse (StereoDrive). Craniotomies (600 μm in diameter) were drilled over the mPFC or PHb, in both hemispheres, according to the following coordinates: mPFC: anterior – posterior (AP) + 1.98 mm from bregma; and medial – lateral (ML) ± 0.3 mm from the midline; PHb: AP −1.8, ML ± 0.82. A third craniotomy (600 μm in diameter) was drilled over the posterior part of the brain, where a ground wire was inserted during the subsequent recording. A titanium headplate, with a 1 cm diameter aperture, was secured to the exposed skull using dental cement, and the headplate aperture sealed with a silicone elastomer (Kwik-Cast) to protect the brain.

### Extracellular in vivo recording in awake mice
Recordings were performed at least three days after a craniotomy, with the mouse placed in a Faraday cage that was darkened to avoid straylight from reaching the mouse's eyes. The mouse was secured to metal arms through its headplate, and covered with a plastic hemi-cylinder to reduce movement. Its feet were placed on a cylindrical treadmill, allowing free walking throughout the recording session. Over the course of the week preceding the first recording session, the mouse was gradually habituated to the restraint with daily sessions of increasing intervals of 5–30 min. Recordings were performed using a 32-site multielectrode array silicon probe with recording sites (area 177 μm², spaced 50 μm) arranged along two columns spanning 750 μm along the probe axis (A1x32-Poly2-5 mm-50s-177, NeuroNexus). The probe, attached to a micromanipulator (IVM Triple, Scientifica) and viewed through a dissecting microscope, was lowered into the brain automatically at a speed of 2 μm/sec to minimize tissue damage[71]. The craniotomy was maintained filled with sterile saline throughout the experiment. Prior dipping of the probe in a red fluorescence dye (1 μg/μL DiI in ethanol) allowed post-hoc identification of the probe placement. To target the different mPFC regions, the probe was inserted to depths of 2700–3300 μm below the dura level. After the probe has reached its final position and before starting a recording session, we allowed the mouse to dark adapt and the neural activity to stabilize for 60 min. Wide-band neural activity (1-8 kHz) was acquired, amplified (x1000), and 16-bit digitized at a 40 kHz sampling rate (OmniPlex, Plexon), and later high-pass filtered at 250 Hz (Offline Sorter, Plexon) for processing of FR. No obvious sleep-inducing effect of light was observed during the recording sessions, possibly due to the uncomfortable posture of the mouse while being head-restrained. Mice appeared to intermittently walk or run on the treadmill, without obvious correlation or anticorrelation to the light stimulus. At the end of a recording session, the probe was retracted, and a similar recording session in the other hemisphere commenced. Then, the craniotomy was closed with a silicone elastomer (Kwik-Cast), and the mouse returned to its home cage. This procedure was repeated for up to 3 days, spanning a week. Recording sessions were performed during the first-half of the dark phase of a 12 h/12 h light/dark cycle [commenced at zeitgeber time 16:42 ± 3:18 (hour:min, mean ± SD) and lasted 45 ± 3.4 min (mean ± SD)]. At the conclusion of the experiment, the mouse was euthanized by cervical translocation immediately following an overdose of isoflurane anaesthesia, and the brain was harvested for later slicing, imaging, identification of the probe position, and neuronal mapping.

### Visual stimulus for in vivo recordings
A recording session included a series of light stimuli at 7 intensities spanning a 6-log change, covering a large part of the operational ranges of rods, cones, and the ipRGCs' photopigment – melanopsin[19,21]. At each light intensity, the stimulus consisted of 20 repetitions of 10 seconds of light followed by 10 seconds of darkness. The light

stimulus, emitted by a white LED (MWWHL4, 3000 K, 570 mW, Thorlabs), was directed toward the mouse eyes through two liquid light guides (5 mm core diameter, LLG05-4H, Thorlabs). Light intensity was modulated using a series of absorptive neutral density filters mounted on a motorized filter wheel (FW212C, Thorlabs) along the light path. Stimulus light intensity (irradiance) was measured using a spectrometer (FLAME-S-UV-VIS-ES, Ocean Optics) coupled with a fibre optic (QP400-1-UV-VIS) and a cosine corrector (CC-3-UV-S), and was calibrated for absolute irradiance (HL-3P-CAL). The measured stimulus irradiance spectrum, together with the spectral absorbance of the mouse lens[72], and the spectral absorbances of mouse rod, cone, and melanopsin pigments[15,73], were used to estimate the quantum catches of the different pigments (9.4–15.4 log photons $cm^{-2} s^{-1}$ for the rod, M-cone, and melanopsin, and 6.6–12.6 log photons $cm^{-2} s^{-1}$ for the S-cone) (Supplementary Fig. 1a, b). These quantum catches corresponded to melanopic equivalent daylight illuminance (melaponic EDI) of 0.005 − 5506 lux across the 7 test light intensities (using the melanopic illuminance for rodents toolbox, https://lucasgroup.lab.manchester.ac.uk/measuringmelanopicilluminance/)[74].    Throughout the study, results were presented with melanopsin quantum catches serving as proxy of light intensity. The variation in melanopsin quantum catches across recording sessions was estimated to equal 0.13 log photons $cm^{-2} s^{-1}$, based on 10 irradiance measurements following repositioning of the LED. To test whether neurons can continuously encode the intensity of light, in a subset of experiments, the 7-intensity recording session was followed by a session comprising 20 repetitions of a bi-phasic stimulus[37]. Log light intensity (log photons $cm^{-2} s^{-1}$) increased linearly and monotonically, spanning a 2-log intensity change, over 30 sec ('increasing phase'), and then linearly and monotonically decreased to the initial intensity over another 30 s period ('decreasing phase'). In both session types, we detected an average delay of 116 ms between the time when a 'turn on' command was sent to the LED driver and the moment when the light actually turned on; this delay was accounted for in all analyses and graphs.

### Mapping neurons in a standard brain

Harvested brains were fixed in 4% paraformaldehyde, and embedded in agarose gel and cut to 60 μm coronal slices with a vibratome (VT1000S, Leica). Fluorescent marks of DiI corresponding to the probe track were imaged using a widefield fluorescence microscope (Olympus BX51, 4x-20x objectives). In three control experiments, the deep edge of the DiI track precisely overlapped with an electrolytic lesion (nanoZ, Plexon) directed specifically at the recording site located closest to the probe tip, confirming the accuracy of the DiI track, as reported previously[75]. Images of brain slices were visualized and manually registered to the Allen Institute Mouse Brain Atlas (Allen Mouse Brain Common Coordinate Framework v3, 10 μm voxel, 2017 release, http://data.cortexlab.net/allenCCF/) using the MATLAB-based tool SHARP-Track[76]. The entire probe track was mapped in a standard mouse brain by annotating 30–40 points along the DiI fluorescent track (made during the recording), typically in several slices, and fitting a line to these points. Next, utilizing the probe's coordinates in a standard brain, the probe's channelmap, and knowledge of the closest neuron to each recording site, we mapped each of the neurons in a standard brain. The resulting coordinates were used to plot the recorded neurons in a 3D brain model using brainrender[77]. For plotting purposes only, jitter (randomly distributed, 0 − 25 microns) was added to each neuron's coordinates to prevent neurons captured through the same recording site from masking one another. Mapping uncertainty was estimated as the standard deviation between the probe track annotations performed by the three experimenters who annotated all the data presented in this study. This yielded an estimated uncertainty of 32, 85, and 39 μm along the rostral-caudal, dorsal-ventral, medial-lateral axes, respectively. To estimate the probability for each neuron to be affiliated with a given mPFC subregion, we created a 3-dimentionl

cloud of 100,000 points distributed normally around the neuron's location by taking the mapping uncertainty calculated above as the standard deviation of the distribution. The neuron's probability to be affiliated with the specific mPFC subregion was estimated as the number of points falling within the original subregion divided by 100,000 points. A similar procedure was employed for estimating the probability for each neuron to be affiliated with a given cortical layer. The incidence of light-responsive neurons (or intensity-encoding neurons) in a given hemisphere was calculated as the number of light-responsive neurons (or intensity-encoding neurons) out of all recorded neurons in that hemisphere. Therefore, the presented estimates of neuron incidences are independent of the placement of the multielectrode array.

### Chemogenetic manipulation and terminal ablation of ipRGC

Opn4[Cre/+] mice were anaesthetized with isoflurane (3% in room air; SomnoSuite), and an AAV inducing expression of inhibitory DREADDs (AAV2-hSyn-DIO-hM4D(Gi)-mCherry; Addgene) was injected (1–2 μl of $10^{13}$ units $ml^{-1}$) into the vitreous humour of both eyes through a glass pipette using a microinjector (Picospritzer III, Science Products GmbH). This led to ipRGCs-specific DREADD/mCherry expression, mainly in the retina's ganglion cell layer, as confirmed by the mCherry labelling. Chemogenetic manipulation of neuronal activity was performed 4 weeks later, through subcutaneous catheter infusion of CNO (5 mg/kg)[78] during in vivo recordings. FR in response to 7 light intensities was recorded twice, first 10 min following saline infusion, and again 10 min following CNO infusion. Reversing the order of saline and CNO infusion was not possible because the CNO effect might linger for several hours[79–81] – longer than the desired duration of an acute recording session. At the completion of the recording, animals were sacrificed, and brain and retinas were harvested for confirmation of DREADD expression.

The effect of CNO-induced inhibition of ipRGCs on locomotion was not assessed. However, no obvious changes in locomotion were observed following saline or CNO infusion. On the other hand, CNO-induced inhibition of ipRGCs leads to pupil dilation[81], which allows a larger amount of light to enter the eye, potentially leading to greater activation of ipRGCs and the light-sensitive mPFC neurons they drive. Indeed, pharmacologically dilating the pupils [1% tropicamide in dimethyl sulfoxide (DMSO), applied on both corneas] slightly increased the FR of light-responsive and intensity-encoding mPFC neurons, albeit not at a statistically-significant level (Supplementary Fig. 9d). In contrast, CNO infusion in our chemogenetic experiments, that also dilated the pupil, decreased the firing rate of mPFC neurons, suggesting that the observed effect of CNO infusion on mPFC firing cannot be explained by an effect of CNO-induced inhibition of ipRGCs on pupil size.

To terminally ablate ipRGCs, Opn4[Cre/+] mice were given bilateral intravitreal injections of an AAV inducing Cre-dependent expression of diphtheria toxin A fragment and constitutive mCherry expression (AAV-mCherry-flex-dtA)[82]. Extracellular in vivo recordings in head-restrained mice commenced 4 weeks following this dtA viral injection. To validate the ipRGCs ablation, retinas were dissected following in vivo recordings and immunostained against melanopsin (primary: dilution 1:600, rabbit anti-melanopsin, ab19306, Abcam; secondary: dilution 1:1000, goat anti-rabbit Alexa Fluor 488, ab150081, Abcam). Fluorescently-marked somata and axons in flatmounted retinas, as well as in mPFC and PHb slices, were imaged using either a fluorescence stereoscope (SMZ25, Nikon) or a confocal microscope (A1R, Nikon; 20x-60x objectives).

### Chemogenetic manipulation of the activity of PHb-to-mPFC neurons

To silence PHb-to-mPFC transmission, WT mice were bilaterally injected with a retrograde Cre/GFP-expressing AAV (AAVrg.hSyn.-HI.eGFP-Cre) in the mPFC's PL/IL, and a Cre-dependent AAV expressing

DREADDs/mCherry (AAV2-hSyn-DIO-hM4D(Gi)-mCherry) in the PHb, leading to selective, albeit not complete, transduction of mPFC-projecting PHb neurons. Chemogenetic manipulation of neural activity was achieved through subcutaneous catheter infusion of CNO (5 mg/kg)[78] 4 weeks later during in vivo recordings, after which animals were sacrificed and the brain harvested for confirmation of DREADD expression. See section *Chemogenetic manipulation and terminal ablation of ipRGC* above for details on the order of saline and CNO infusion. Validating the CNO-induced inhibition of PHb neurons was not attempted because FR reduction induced by CNO could not be distinguished from FR reduction due to minute movements of the multi electrode array following CNO injection, which often occur during recordings, and of which available spike sorting routines cannot handle.

Chemogenetic manipulation experiments involved 2 to 4 viral intraocular/intracranial injections in each mouse. When attempting such a large number of injections, the probability of accurately injecting into all targets is low. AAV expression also varies across injections and mice. Nonetheless, experiments, in which we achieved accurate targeting and optimal viral expression, were successful and delivered consistent results.

## Optogenetic inhibition of axon terminals of mPFC-projecting PHb neurons

To photoactivate eOPN3, we used a 50 mW, 532 nm diode-pumped solid-state (DPSS) green laser (Shanghai Laser & Optics Century), which was adjusted to deliver 7.5 mW mm$^{-2}$ at the tip of the optotrode (ASSY-37 H7b, 32 sites, 1 shank, 9 mm length; fibre 200 μm diameter, N.A. 0.39; Cambridge Neurotech). This laser power was selected to ensure maximal activation of eOPN3, while accounting for its power-response function and spectral sensitivity[34]. To minimize light contamination reaching the eyes from the green laser light, which could potentially stimulate the investigated light-sensitive pathways, we employed a photoactivation protocol that obviates simultaneous eOPN3 activation with recordings of light-evoked FR in mPFC neurons. Thanks to a long-lasting effect of light on eOPN3-induced inhibition (time for 50% recovery is 6.6 min[34]), the photoactivation protocol comprised a train of 30 s pulses at 2.33 min intervals (once every 7 repetitions of the light stimulus), ensuring that the recorded mPFC firing was only in response to the white light stimulus.

In addition to green laser light, an unknown fraction of the white stimulus light likely penetrated the brain through the cranium and/or craniotomy, and reached the PHb and its targets. The highest intensity of white light that could have penetrated the brain through the cranium was measured as 0.123 mW mm$^{-2}$, and the highest intensity that could have fallen on the craniotomy was 0.027 mW mm$^{-2}$, both above the power shown to induce eOPN3 activation[34]. However, because the green laser light already activated eOPN3 to the maximum, white light penetrating the brain could not activate eOPN3 in the axon terminals innervating the mPFC any further. Nonetheless, we cannot exclude the possibility of some eOPN3 activation in non-mPFC-projecting PHb neurons.

Since both eOPN3 and the pathways underlying mPFC photosensitivity are modulated by light, we did not compare light-evoked mPFC FR before vs. during eOPN3 photoactivation. Instead, we recorded light-evoked mPFC FR under sustained eOPN3-induced inhibition of axon terminals of mPFC-projecting PHb neurons. This approach is superior to terminal ablation of PHb neurons, e.g., via diphtheria toxin A (dtA), which could have long-term detrimental effects on physiology and behaviour, especially when considering the heavy PHb projections to the mPFC, NAc, caudate putamen, zona inserta, and thalamic reticular nucleus[10,38]. For control, we used the same optotrode and photoactivation/recording protocol in mice in which instead of eOPN3, the fluorescent reporter mScarlet was virally expressed in PHb neurons.

## Analysis of in vivo electrophysiological data

**Identification of single neurons (units).** Spike sorting, i.e., isolating the responses of single neurons, was performed using the software Kilosort 3 https://github.com/MouseLand/Kilosort, followed by curation with the software Phy 2 https://github.com/cortex-lab/phy[83]. Clusters identified by Kilosort 3 were taken to be single neurons if they exhibited: (1) a uniform typical spike waveform shape, (2) a significant refractory period as determined by spike autocorrelation and inter spike intervals (ISI), (3) a FR higher than 0.5 Hz, and (4) a normal distribution of spikes' amplitudes. For each isolated neuron and for each of the 20 stimulus presentations, the spike times over a 20-sec period composed of the 10-sec stimulus duration, the 4 sec preceding the light onset and the 6 s following light offset, were extracted. These spike trains were binned at 100 ms to construct peri-stimulus time histograms (PSTHs). For each neuron, we calculated 7 × 20 PSTHs corresponding to 20 stimulus presentations over 7 intensities. Each PSTH was baseline-normalized by subtracting the FR at each bin from the baseline FR, taken as the mean FR across the 3 s preceding light onset.

**Transient and persistent light-responsiveness classification.** Early on, we observed two consistent patterns in FR changes following light onset: transient, and persistent. Considering the potentially-different origins of the two patterns and the roles the two may play in light-signals processing and in intensity encoding, we aimed to identify neurons that transiently and/or persistently respond to light. For transient responses, we considered the time-average FR (abbreviated hereafter 'FR') during the first 1 sec of the stimulus ('early' window, $FR_1$), while to identify neurons that respond to light also or only persistently, we considered the time-average FR during the last 6 s of the 10-sec long stimulus ('steady-state' window, $FR_2$). To classify neurons as light-responsive vs. light-nonresponsive, we performed a paired two-sided permutation t-test on the FR before and during stimulus (a time window spanning the 3 sec preceding the stimulus, 'baseline' window, $FR_0$ vs. $FR_1$ or $FR_2$ for 'transient' or 'persistent'). For this test, all FR records in response to the three highest stimulus intensities were pooled, a total of 60 FR records per neuron (20 repetitions x 3 intensities). To minimize the influence of other variation sources on subsequent analysis, and considering that the mPFC is involved in a large array of functions and is heavily modulated by affect and cognition[84,85], potentially introducing diverse sources of variation to the recorded neuronal activity, prior to testing light-responsiveness we identified and omitted up to 3 outlier responses (out of the 20 repetitions for each intensity) in the 'early' and 'steady-state' window data (MATLAB 'isoutlier' function with the 'median' method was used).

**Identification of light-intensity-encoding neurons.** To evaluate a neuron's ability to persistently encode the intensity of light, for each neuron we constructed an intensity-response curve based on the mean FRs in response to the 7 stimulus light intensities during the 'steady-state' window ($R$), and performed a non-linear regression of the data with the sigmoidal Naka-Rashton function[86]:

$$R = R_{max} * 10^{(nE)}/(10^{(nE)} + 10^{(nK)}),  \quad (1)$$

where $R_{max}$ is the neuron's predicted maximum response, $n$ the slope, $E$ the light intensity (expressed as log photons cm$^{-2}$ s$^{-1}$), and $K$ the neuron's sensitivity. Root mean square error (RMSE) from the fitted sigmoid was calculated and compared to the RMSE null distribution (arrived at by repeated shuffling of the responses along the intensity (x) axis and fitting the shuffled data to a sigmoid; 100 permutations yielded a 0.01 probability resolution). Neurons for which the RMSE was smaller than the 5$^{th}$ percentile of the RMSE null distribution were classified as intensity-encoding.

**Classification of neurons by light-evoked response type.** To identify response (functional) types of persistent light-intensity-encoding neurons, we performed a principal component analysis (PCA) on the mean PSTH for the highest intensity (15.4 log photons $cm^{-2}$ $s^{-1}$; the PSTH was binned into 0.2 s segments to minimize noise), followed by unsupervised clustering (Gaussian Mixture Models) on the PCs scores, which accounted for 85% of the variance. This clustering method fits multiple gaussian models to the data, and subsequently clusters the neurons according to the probability of each neuron to overlap with the distribution of the various gaussian models. The model that yielded the minimum value of the Akaike's Information Criterion (AIC) was chosen as the model with the optimal number of clusters[87].

**Classification of mPFC neurons into excitatory and inhibitory cells.** To identify 'fast-spiking' interneurons and 'regular-spiking' pyramidal neurons, we calculated four commonly used metrics of the extracted mean action potential waveform of each single neuron[88–90]. These metrics were: spontaneous FR, trough to peak duration, trough to peak ratio, and trough half duration. We then subjected the four normalized (0-1) metrics to PCA and performed hierarchical clustering on the PCs accounting for 95% of the variance (cluster function, MATLAB), to classify the neurons into 2 clusters, matching previous classification features of putative fast spiking and regular spiking neurons.

**Light-evoked firing rate change relative to baseline.** The significance of the effect of light on neuronal firing depends on the neuron's (spontaneous) baseline FR. For example, let us consider two neurons that differ in baseline FR, but upon light exposure, increase their firing by the same number of spikes per sec. The change in FR relative to the baseline firing would be greater for the neuron with the lower baseline FR. The relative change of a neuron's light-evoked activity was calculated as the difference in FR between the activity in response to the highest stimulus intensity and the baseline, normalized by the baseline.

$$M = 100 \times \frac{\left( FR_{1/2} - FR_0 \right)}{FR_0} \qquad (2)$$

where $FR_0$ is the baseline FR, and $FR_1$ and $FR_2$ the FR during the 'early' or 'steady-state' window, respectively. A relative change of 0 represents no change in FR, while a positive or negative change corresponds to an increase or decrease in FR relative to baseline, respectively.

**Absolute firing rate as a function of light intensity.** For selected analyses, we calculated the steady-state FR across intensities, averaged across either all identified neurons, all light-responsive neurons, or all intensity-encoding neurons. To enable averaging the responses of neurons whose light-evoked responses increase with those whose light-evoked responses decrease with light intensity, we negated the responses of neurons whose FR decreased with increasing light intensity.

*Latency.* Latency to the late response component. For each neuron, latency was taken as the preceding time point closest to the onset of the late response component (sought across the 10 s stimulus) when FR exceeded 1 standard deviation of the baseline FR. Mean ± s.d. latency per response type were then calculated.

**Latency to ON peak.** Latency was calculated using the same method as above. However, the peak was sought across the first 2 s of a mean PSTH calculated at higher temporal resolution (10 ms), for each response type.

**Decay time.** Decay time was taken as the time following the stimulus offset when the mean FR of each neuron in response to the highest stimulus intensity returned to the neuron's baseline FR.

## Patch-clamp whole-cell recordings in ipRGCs
To confirm the inhibiting effect of CNO on ipRGC photosensitivity, we utilized whole-cell current-clamp recordings from DREADD/mCherry-positive ipRGCs in flat-mounted retinas of Opn4$^{Cre/+}$ mouse. See section 'Chemogenetic manipulation and terminal ablation of ipRGC' for details on intravitreal injections for chemogenetic manipulation.

### Retinal dissection
Mice were euthanised with a lethal dose of $CO_2$ followed by cervical translocation. Eyes were removed and immersed in oxygenated Ames medium (95% $O_2$, 5% $CO_2$; Sigma-Aldrich; supplemented with 23 mM $NaHCO_3$ and 10 mM D-glucose). Under dim red light, the globe was cut along the ora serrata, and cornea, lens and vitreous removed. Four radial relieving cuts were made in the eyecup. The retina was flat-mounted on a custom-machined hydrophilic polytetrafluoroethylene membrane (cell culture inserts, Millicell[28]) using gentle suction, and secured in a chamber on the microscope stage. Retinas were continuously superfused with oxygenated Ames' medium (32–34 °C).

### Whole-cell patch-clamp electrophysiology
Current-clamp recordings of isolated flat-mount retinas were performed using a Multiclamp 700B amplifier, Digidata 1550 digitizer, and pClamp 10.5 data acquisition software (Molecular Devices; 10 kHz sampling). Pipettes were pulled from thick-walled borosilicate tubing (P-97; Sutter Instruments). Tip resistances were 4–8 MΩ when filled with intracellular solution, which contained (in mM): 120 K-gluconate, 5 NaCl, 4 KCl, 2 EGTA, 10 HEPES, 4 ATP-Mg, 7 phospho-creatine-Tris, and 0.3 GTP-Tris (pH 7.3, 270–280 mOsm). Red fluorescent dye (Alexa Fluor 568; Invitrogen) was added to the intracellular solution for visual guidance during two-photon imaging and intracellular dye-filling. Light-evoked FR of ipRGCs was recorded in response to 7–10 stimulus light intensities, first while the retina was superfused with oxygenated Ames' medium, and again, 30 min after supplementing the Ames' medium with CNO (0.3 μg CNO / 1 ml Ames' medium). Following recording, filled cells were documented through two-photon Z-stacks.

### Visual stimulation
Stimuli were generated using a light beam from a 405 nm LED (M405LP1, Thorlabs), transmitted through the microscope's substage optics, projected onto the retina using a set of lenses, and focused onto the photoreceptor outer segments as a uniform centre spot (150 μm in diameter) on a dark background. Incorporating a motorized filter wheel (FW212C, Thorlabs), mounted with 10 reflective neutral density (ND) filters (Thorlabs), enabled varying the stimulus intensity, which was measured as described above. The measured stimulus irradiance spectrum, together with the spectral absorbances of mouse rod, cone, and melanopsin pigments[15,73], were used to estimate the quantum catches of the different pigments (8.6–13.6 log photons $cm^{-2}$ $s^{-1}$ for melanopsin; 8.3–13.3 log photons $cm^{-2}$ $s^{-1}$ for the rod and M-cone; and 8.1–13.1 log photons $cm^{-2}$ $s^{-1}$ for the S-cone). Photoisomerization rates were derived from the pigments' quantum catches while accounting for the estimated rod (0.85 μm$^2$) and cone (1 μm$^2$) collecting areas[32,91]. Rates ranged -10$^0$–10$^5$ photoisomerizations/s (R*/photoreceptor/s) for rods, cones, and melanopsin[92,93]. The light stimulator was controlled by custom software using Psychophysics Toolbox under MATLAB (The MathWorks). For analysis, voltage traces were down-sampled to 0.1 kHz, and the response amplitude was taken as the lower envelope of the voltage response. Dynamic range (DNR) was estimated as the 10$^{th}$ and 90$^{th}$ percentiles of the first derivative of the fitted sigmoid.

## Statistical analysis

**Continuous data.** As continuous data (e.g., FR, latency, decay time) often deviated from assumptions of normality (Kolmogorov Smirnov test) and homoscedasticity (Bartlett's test), we utilized appropriate permutation tests to determine the significance of differences between multiple samples, at p 0.05. We used either a permutation t-test for two independent samples, a permutation t-test for paired samples, or a one-way permutation analysis of variance (ANOVA). post-hoc to ANOVA, we performed pair-wise comparisons using a permutation t-test for two independent samples, while correcting for multiple comparisons using the 'tmax' method[94,95].

**Discrete data.** When comparing the incidence of light-responsive or intensity-encoding neurons between brain hemispheres, brain regions (e.g., mPFC vs. MOs) or before/after manipulations, we tested for statistical significance per recording session, typically using a permutation t-test or a 1-way ANOVA. This approach, however, is limited when applied to comparisons of incidence of light-responsive or intensity-encoding neurons between subregions and cortical layers, because of an unavoidable potential screening bias caused by the physical size of the multi-electrode. The multi-electrode array, with its 32 electrode sites spanning $750\,\mu$m, traverses multiple subregions and layers in each session, with the covered proportion out of a subregion's full size varying between subregions. For example, all sessions targeting the ventral mPFC included only a small portion of the AC, which constitutes the dorsal mPFC. Consistent with the sampling limitation, in all these sessions the incidence of light-responsive and intensity-encoding neurons in the AC was low (Supplementary Fig. 1l, m). To reduce our sensitivity to the structural uneven sampling, for comparing the incidence of light-responsive or intensity-encoding neurons between subregions and layers we calculated the neurons' incidence as their total number across all recording sessions in a given subregion or layer divided by the total number of the neurons identified in that locality across all sessions. To identify statistically-significant differences between two distributions of incidences, the non-parametric $\chi^2$ (Chi-square) test was then utilized.

To establish the minimum detectable effect size in $\chi^2$ tests, we conducted a prior sensitivity analysis using the G*Power 3 program (version 3.1.9.4)[96] with power (1-β) of 0.8 and an α error rate of 0.05. For comparison of the incidences of intensity-encoding neurons in the IL/PL vs. the remaining mPFC subregions ($n = 1675$, df = 1), the sensitivity analysis indicated a capability for identifying small effect sizes ($\varphi > 0.068$). For comparison of the distribution of intensity-encoding neuronal types between hemispheres ($n = 252$, df = 3), the sensitivity analysis indicated a capability for identifying medium effect sizes (Cramer's V > 0.208). For testing the effect of chemogenetic inhibition of mPFC-projecting PHb neurons on the incidence of light-responsive neurons (df = 1), sensitivity analysis indicated a capability for identifying medium/large effect sizes ($\varphi > 0.317$) in the DREADD-expressing mice ($n = 78$), and small/medium effect sizes ($\varphi > 0.164$) in the mCherry-expressing mice ($n = 290$). For comparison of the distribution of intensity-encoding neuronal types between the PHb and mPFC ($n = 306$, df = 3), sensitivity analysis indicated a capability for identifying small/medium effect sizes (Cramer's V > 0.189).

Since sensitivity analysis approaches for permutation tests are debatable[97], we report the effect size for all the presented comparisons. The appropriate effect size statistic was determined for each type of statistical test[98], and was calculated using the 'measures of effect size (MES) toolbox' (https://github.com/hhentschke/measures-of-effect-size-toolbox). We used the Phi ($\varphi$) statistic for $2 \times 2$ $\chi^2$ tests, Cramer's V statistic for $2 \times 5$ $\chi^2$ tests, eta-square ($\eta^2$) statistic for 1-way ANOVA, Cohen's $d$ unpaired statistic for premutation t-tests for two independent samples, and Cohen's $d$ paired statistic for premutation t-tests for paired samples. All analyses were performed in Matlab.

## Reporting summary

Further information on research design is available in the Nature Portfolio Reporting Summary linked to this article.

## Data availability

The data generated in this study have been deposited in Figshare: "Prefrontal cortex neurons encode ambient light intensity differentially across regions and layers", under accession code: https://doi.org/10.6084/m9.figshare.23659974 [https://figshare.com/s/cf9dd54f122789dcd4a5]. Source data are provided with this paper.

## Code availability

The code generated during this study is available on https://github.com/elyashivzangen/light-and-the-mPFC-code (https://doi.org/10.5281/zenodo.11265775)[99].

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

## Acknowledgements

We thank David Berson for kindly providing us with Opn4$^{Cre/+}$ mice. We are thankful to our colleagues who provided invaluable theoretical, data-analytic and technical advice, Avihu Klar, Yonatan Kupchik, Alex Binshtok, Ariel Gilad, Dan Rokni and Eli Shmueli. This project was supported by the Brain & Behavior Research Foundation (BBRF) grant (#30072), Israel Science Foundation (ISF) grant (#1134/21), the National Institute for Psychobiology in Israel grant (228-19-20b), and Institute for Medical Research Israel-Canada (IMRIC) Center for Addiction Research (ICARe) grant, awarded to S.S.

## Author contributions

S.S., E.Z., and S.H. designed the study and developed the theoretical framework. M.O. performed in vivo electrophysiological recordings, chemogenetic manipulation experiments, and histology. E.Z., S.H., O.A., E.Z., N.S., D.-C.-H., S.N., and Y.L. performed in vivo electrophysiological recordings. E.Z. performed optogenetic manipulation experiments. C.L.

performed all in vitro electrophysiological recordings. I.D., E.H., H.R., and C.L. performed intravitreal and intracranial injections. Y.S. performed immunostaining and imaging of retinas. E.Z., S.H., and S.S. analysed all morphological and electrophysiological data. S.S. and E.Z. wrote the paper.

## Competing interests

The authors declare no competing interests.
