## [Peer Review File · Nature Communications]

Prefrontal cortex neurons encode ambient light intensity differentially across regions and layersReviewers' Comments:

Reviewer #1:

Remarks to the Author:

The authors show for the first time that neurons in the medial prefrontal cortex (mPFC) can encode light by showing photosensitivity. Mechanistically, they showed that knocking out intrinsically photosensitive retinal ganglion cells (ipRGCs) or the perihabenular nucleus (PHb) reduced photosensitivity in the mPFC. They were able to distinguish four types of light intensity-encoding neurons, two with increased firing rate in response to increasing light intensity and two vice versa, not only in the mPFC but also in the PHb with even higher sensitivity and shorter latency. In addition, they found evidence that light had a dual effect on limbic activity: suppressing the prelimbic and enhancing the infralimbic. They interpreted this as the opposing roles of these brain regions in fear conditioning, drug seeking and anxiety. I would like to congratulate the authors for their novel and original work, which has translational potential to be studied in humans and potentially for clinical applications.

Could you please provide more details about the light used in the experiments, its spectral composition, and how it's likely to affect the different photoreceptors?

Could you have a better description of which types of ipRGCs were silenced?

Was there a specific subtype responsible for the light effects in the mPFC?

Can you better emphasise/explain the comparison with your own work published in 2022 in humans?

Now that you have looked at the mechanisms?

Can you give a trajectory or outlook for the future? What is next and what can potentially be translated into human and even clinical settings?

Reviewer #2:

Remarks to the Author:

I enjoyed reading the paper of Sabbah et al. It is a timely research that provided the neuronal circuitry that underlies previous observations of the impact of light on cortical activity in humans. The paper is clearly written and I don't have major issues with the methods and interpretation of the results and will raise what I consider to be important and minor issues.

Important

- A dedicated statistical analysis section should be provided including how the animal, neurons and repeated measures were dealt with, correction for multiple comparisons, together with a power or sensitivity analyses. Also, how was the sex of the animals taken into account in the statistical analyses.

- In addition to the number of neurons in which light responses were detected, the number of animals in which it was detected should be reported for all analyses (together with the mean \pm SD of neurons per animal). This is not always done and will allow the full appreciation of the generalisation of the findings. Likewise, figures should make clear whether the n corresponds to the numbers of neurons or of animals. Likewise supplementary information should detail the number of sessions and neurons per mouse in addition to the number of males and females.

- An important limitation of the study is the lack of behavioural measures included. We do not know to which aspect of behaviour the different cell types contribute to and whether they do contribute to any behaviour. This limitation should be clear. Likewise, translation from animal research to humans is not straightforward (e.g. Braak 2015 Brain) and remains to be done such that the authors should be cautious in their statement throughout the text.

- It would be good if figure 6 was to make postulates on which cell type underlies which behaviour. Still on Fig 6, projection f and g are not fully clear (f gets mixed up with l and it's unclear whether g also takes part of f when it crosses it). I recommend using different colours

- Section "Light intensity continuously modulates mPFC neuronal activity" relies on only 3 animals given the small number of mice included

- I wonder whether the lateralisation of some effects could be related to difference in the location of the probes. It should be commented in the text.
- The author should provide detailed characteristic of the light conditions they used (at least in mel EDI lux) and mention what illuminance they intended to administer and the potential deviations. The precise time of day of the experiments should be provided.

Minor

- Reference to human studies studying light and mood and/or emotion should be made to strengthen the rationale. Likewise, interpretation would be strengthened by the inclusion of more human studies finding mPFC modulation by light.
- Environmental light detection relies on slow time scale and this comes at odds with the use of the term "flashes" which should be replaced by "pulses", "blocks" or "exposure"
- The abstract makes the appealing statement that results could it be useful for therapies. However the paper makes no clear statement regarding this aspect.
- For some figures, the meaning of * is not provided in the legend
- Figure display of brain images should specify left and right
- The 3rd sentence of the intro is very long and should be broken in smaller pieces for clarity.
- Figure 3a, a scale for firing rate should be provided to each panel
- In what phase of the menstrual cycle were the tested females?
- Given the masking / sleep-inducing effect of light, the authors should mention mouse behaviour during the recordings at least qualitatively.

Reviewer #3:

Remarks to the Author:

In this study, the authors investigate light responses and irradiance sensitivity in the mPFC in awake, head-fixed, mice - finding a surprisingly high proportion of cells respond to light across its various subdivisions including a subset that show monotonic changes in firing as a function of irradiance. They also find similar types of responses in the PHb and provide some evidence that a pathway involving ipRGCs and mPFC-projecting PHb neurons might mediate such effects using chemogenetic manipulations and ipRGC ablation.

Overall I find the study interesting. Characterisation of light response types and their anatomical locations is carefully performed and will provide a valuable addition to our understanding of how light influences brain function. This is somewhat let down by the fact that much of the description around anatomical localisation is descriptive without any statistical analysis upon which to assess which differences are meaningful and which aren't. That issue should be easy to address, however. A more substantial and harder to address issue relates to the circuit mapping aspects of the study:

- 1) Given that these are awake/behaving animals can the authors exclude the possibility that light responses observed are secondary to some general behavioural or physiological responses to light rather than specific pathways carrying light information (eg ipRGC->PHb->mPFC)
- 2) For chemogenetic studies such as those employed here it is not sufficient to rely on comparisons between CNO and saline. Given that it is now widely accepted that CNO can exert off-target (non-DREADD-mediated) effects, really these studies need a non-DREADD vector control group (eg mCherry) that also receives CNO. Note also that following point 1, above, the DREADD manipulations if proved effective could still be acting via indirect behavioural and/or physiological effects (eg changes in running speed, pupil size etc.) which would need to be controlled for.

Leaving those points aside, while I agree the data are potentially consistent with some effect of inhibiting/lesioning ipRGCs (and to a lesser extent also inhibiting mPFC-projecting PHb cells), the effects are not especially compelling. This may be due to low n numbers and/or low penetrance of the

manipulations. Either way, I think it would be hard to be certain from the present data that ipRGC->PHb is the primary circuit that mediates effects of light/irradiance on the mPFC. Indeed it is notable that effects of light on the mPFC are substantially slower than those in the PHb (40ms as reported here). That seems like quite a long delay for a monosynaptic connection. The study would be substantially enhanced, if data specifically linking the PHb projection to light induced changes in the mPFC could be strengthened and/or data on the physiological relevance of mPFC light responses could be provided.

Specific comments:

P4: 'The incidence of transient and persistent light-responsive neurons was highest in the AC, PL, and IL (transient: 43-49%, persistent: 34-42%, of identified neurons) (Fig. 1f)' – do authors find evidence that incidence varies significantly as a function of subregion? In the absence of any statistical test to exclude the alternate possibility that this just reflects a random sampling artefact this sentence should be removed or revisited.

P4: 'The incidence of light responsive neurons also varied across hemispheres, being larger in the left hemisphere (46% transient and 39% persistent in the left hemisphere, vs. 38% transient and 32% persistent in the right hemisphere) (Fig. 1g)' –as above, is this actually a meaningful difference?

P4: 'An important characteristic of a system capable of mediating an effect of light on emotion and cognition would be the ability to persistently encode the intensity of light.' – I This is a bit circular, effects of 'light' on emotion and cognition needn't necessarily be restricted to modulation according to ambient light intensity.

P5: 'Persistent intensity-encoding neurons were found throughout the mPFC (Fig. 2i), with a slightly higher incidence in the left hemisphere (17%, vs. 12% in the right hemisphere). The regions with the highest incidence of these neurons were the PL and the IL' – Same comments as above apply to much of this whole paragraph. A lot of qualitative statements are made which might simply reflect random sampling effects rather than meaningful differences, especially given the low numbers of cells in each class.

P5: 'Analysing the (darkness) spontaneous firing and action potential waveforms revealed that the four types corresponded to mixtures of presumptive pyramidal neurons and interneurons' – I think this statement is a bit strong. I find the analysis in extended figure 2 convincing that this is more than one group (perhaps 3 based on panel d?) and one would assume the main group is pyramidal neurons. Nonetheless, given it is hard to be certain about this classification, I think suggested would be better than 'revealed'

P6, 1st paragraph – again many qualitative statements that are hard to evaluate without any statistical analysis.

P7- 'Dependence of mPFC photosensitivity on input from the presynaptic PHb' section – I find this section very hard to follow. Fig 4a starts with the experimental paradigm for viral labelling and example image of transduced neurons. The rest of the data shown in this figure, so far as I can gather, compares the properties of PHb and mPFC cells but has nothing to do with the DREADD manipulation per se. There is some data on effects of the DREADD manipulation of PHb on mPFC in extended data fig 7 but that is hard to evaluate and is not especially convincing as currently presented, nor is the paradigm explained in sufficient detail. Are the same cells tested both under saline and CNO? If so is the order of testing randomised? Did the authors validate whether the approach actually silenced PHb neurons? In terms of the data presented, there is information on % cells under saline vs. CNO but actual cell numbers are unclear, and there is no statistical analysis reported. Given the low n numbers for subsequent panels (e.g; n=9 and n=3 respectively), I suspect the percentages shown in panels a-d must come from a very low overall number of cells and so I

wonder whether statistical tests would reveal any meaningful difference. The average response profiles shown in panels e and f suggest maybe a modest reduction for suppressed cells (e), while enhanced responses (f) are so weak under baseline conditions it is hard to say whether there is any meaningful effect at all.

P8: 'Analysing the spontaneous firing activity and action potential waveforms of identified PHb neurons again yielded two distinct neuronal populations' – the analysis shown does not, to me, convince that there are two distinct populations (as opposed to a single population with a long tail), nor do I think it valid to assume that cells below an arbitrary cutoff are inhibitory. As a more general point, the use of spike shape to identify inhibitory interneurons may have validity for cortex and has (as the authors note) also been used for other brain regions. I don't believe the reliability of this approach has ever actually been validated for subcortical regions however. As such the section of the manuscript and associated data figure is highly speculative.

Fig 2: axes on k-m are misleading. If I understand what is being shown correctly, for k it would be appropriate to use % identified neurons (as in Figure 1). For l-m it should be % of irradiance coding neurons (or similar). Indeed for the latter that actual percentages of all identified cells are very small – in that regard it might be more informative to show the raw proportions.

RESPONSE TO REVIEWER COMMENTS

Reviewer #1 (Remarks to the Author):

The authors show for the first time that neurons in the medial prefrontal cortex (mPFC) can encode light by showing photosensitivity. Mechanistically, they showed that knocking out intrinsically photosensitive retinal ganglion cells (ipRGCs) or the perihabenular nucleus (PHb) reduced photosensitivity in the mPFC. They were able to distinguish four types of light intensity-encoding neurons, two with increased firing rate in response to increasing light intensity and two vice versa, not only in the mPFC but also in the PHb with even higher sensitivity and shorter latency. In addition, they found evidence that light had a dual effect on limbic activity: suppressing the prelimbic and enhancing the infralimbic. They interpreted this as the opposing roles of these brain regions in fear conditioning, drug seeking and anxiety. I would like to congratulate the authors for their novel and original work, which has translational potential to be studied in humans and potentially for clinical applications.

Reviewer 1 comment 1:

Could you please provide more details about the light used in the experiments, its spectral composition, and how it's likely to affect the different photoreceptors?

We thank the reviewer for this comment. We generated new plots showing the light stimulus spectrum, lens transmission, and absorbance of photoreceptors. To estimate how the light stimulus affected the different photoreceptors, we also generated a plot showing the quantum catches of each pigment, at the lowest and highest stimulus intensities. These new plots were added to Extended Data. Fig. 1. A statement describing this information has been added to the Methods section (line 579-583).

Reviewer 1 comment 2:

Could you have a better description of which types of ipRGCs were silenced? Was there a specific subtype responsible for the light effects in the mPFC?

Previous retrograde transsynaptic tracing showed that ipRGCs feed the PHb-mPFC pathway, and retrograde tracing showed that the PHb is innervated by the M1 and M4 ipRGC types. However, to the best of our knowledge, there is no proven way to selectively inhibit any ipRGC type for *in vivo* experiments. Therefore, at this time, the contribution of specific ipRGC types to mPFC photosensitivity is unknown. A similar statement has been added to the discussion section (lines 395-398).

While it is impossible to determine which ipRGC types have been silenced during the *in vivo* chemogenetic experiments, it is safe to assume that DREADD-expressing ipRGCs have been silenced. Thus, to determine the identity of DREADD/mCherry-expressing ipRGCs, we extracted and imaged the retinas of mice upon completion of the chemogenetic experiments. DREADD/mCherry expression was evident in the OFF sublamina of the inner plexiform layer (IPL), where M1, M3, and M6 ipRGCs stratify, as

well as in the ON IPL, where M2, M3, M4, M5 and M6 ipRGCs stratify. The relatively dense OFF IPL plexus suggests that labelled neurites include dendrites of M6 cells in addition to dendrites of the sparsely-branched M1 and M3 cells. Moreover, the diverse soma sizes encountered in the ganglion cell layer (GCL), ranging from ~10 μm (M1) to ~25 μm (M4), suggests the expression of DREADD in multiple ipRGC types, in addition to M1 and M4. This information has been added to the caption of Extended Data Fig. 8.

Reviewer 1 comment 4:

Can you better emphasise/explain the comparison with your own work published in 2022 in humans? Now that you have looked at the mechanisms?

The homology between the rodent and human mPFC is debatable, yet the rodent PL and IL are generally thought to correspond with the human pregenual AC and the subgenual AC, respectively. Accordingly, light-induced suppression of net activation in the mouse PL would be consistent with the activity suppression we reported in the equivalent human pregenual AC. Light-induced enhancement of net activation in the mouse IL however, is inconsistent with the activity suppression reported in the equivalent human subgenual AC. Moreover, the time course of sustained light-evoked human PFC responses, and their susceptibility to prior light exposure, resembled those of ipRGCs. Thus, as revealed here in the mouse, intensity-encoding in the human PFC might also be shaped by ipRGC drive. While ipRGC signalling might be transmitted to the mPFC through the visual cortex, central amygdala, lateral hypothalamus, and PHb, the latter has never been reported in humans. If a human PHb-analog exists, a probable location would be bounded by the dentate gyrus, lateral habenula, and the central-lateral nucleus of the thalamus, where it is positioned in mice. BOLD 7-Tesla functional MRI, which is ideal for the detection of small brain nuclei, might allow exploring the existence of a PHb analog in humans. A similar statement has been added to the discussion section (lines 457-483).

Reviewer 1 comment 5:

Can you give a trajectory or outlook for the future? What is next and what can potentially be translated into human and even clinical settings?

Translation from animal research to humans is non-trivial. Nonetheless, our results, together with prior observations in mice implicating a PHb-to-mPFC/NAc transmission in shaping the influence of light on depression-like behaviours, offer a functional link between light exposure and mPFC-mediated affective phenomena in mice, and complement our findings in humans. An intriguing avenue would be harnessing the new knowledge of the characteristics of mPFC light responsiveness to develop therapies for mPFC-modulated emotional and cognitive processes and behaviours. For example, our demonstration that light suppresses PL activity but enhances IL activity, mirrors these mPFC regions' contrasting roles in fear-conditioning, drug-seeking, and anxiety. This suggests the potential of carefully-designed light therapy protocols in alleviating anxiety and addiction disorders. Moreover, uncovering the neural mechanisms underlying mPFC light responsiveness might be utilized for the enhancement of cognitive performance through the design of optimal lighting environments. Yet, it remains to be

clarified whether a PHb-analog exist in human, and to what extent light-evoked responses in nocturnal species as the mouse recapitulate those encountered in the diurnal human. These new statements have been added at the end of the discussion section (lines 484-497).

Reviewer #2 (Remarks to the Author):

I enjoyed reading the paper of Sabbah et al. It is a timely research that provided the neuronal circuitry that underlies previous observations of the impact of light on cortical activity in humans. The paper is clearly written and I don't have major issues with the methods and interpretation of the results and will raise what I consider to be important and minor issues.

Important

Reviewer 2 comment 1:

- A dedicated statistical analysis section should be provided including how the animal, neurons and repeated measures were dealt with, correction for multiple comparisons, together with a power or sensitivity analyses.

We thank the reviewer for this important comment. We have now added a 'statistical analysis' section (lines 851-897), in which we describe in detail the statistical tests used for discrete and continuous data.

The majority of data reported in this study often deviated from assumptions of normality and homoscedasticity. Therefore, to determine the significance of differences between samples, we utilized appropriate permutation tests. We used either a permutation t-test for two independent samples, a permutation t-test for paired samples, or a one-way permutation analysis of variance (ANOVA). Post-hoc to ANOVA, we performed pairwise comparisons using a permutation t-test for two independent samples, while correcting for multiple comparisons using the 'tmax' method. The approach of using permutation tests, however, was found to be non-optimal for testing for differences in the incidence of light-responsive or intensity-encoding neurons in a given mPFC subregion or cortical layer (see more details in the statistical analysis section). In those cases, to determine whether there is a statistically-significant difference between two distributions of incidences (across mPFC subregions or across cortical layers), the non-parametric χ^2 (Chi-square) test was utilized.

Following the reviewer's comment, we performed sensitivity analysis for the comparisons in which the non-parametric χ^2 (Chi-square) test was used. The results of these sensitivity analyses were added to the Statistical analysis sections under Methods. Several recent studies suggested approaches for performing sensitivity analysis/power analysis for permutation tests using Monte Carlo algorithms, yet these approaches have never been validated. Therefore, we opted to calculate effect size for all the comparisons throughout the main manuscript and figures, and we are now reporting these effect size estimates for each test. The appropriate effect size statistic was determined for each type of statistical test, and calculated using the measures of effect size (MES) toolbox (<https://github.com/hhentschke/measures-of-effect-size-toolbox>). We used the Phi (ϕ) statistic for 2 x 2 χ^2 tests, Cramer's V statistic for 2 x 5 χ^2 tests, eta-square (η^2) statistic for 1-way ANOVA, Cohen's d unpaired statistic for

permutation t-tests for two independent samples, and Cohen's *d* paired statistic for permutation t-tests for paired samples.

Reviewer 2 comment 2:

Also, how was the sex of the animals taken into account in the statistical analyses.

Females and males were used interchangeably, however, due to misunderstanding between the lab personnel performing the experiments, sex was not documented in tangent with recordings that constitute the main data set. To probe for potential sex differences in mPFC photosensitivity, we performed an additional, limited, sex-controlled set of experiments. Focusing on the PL and IL regions, that are innervated by the PHb, we recorded in 3 females and 3 males during 8 (F) and 5 (M) recording sessions. The sex-documented data set included 301 (F) and 133 (M) neurons, of which 70 (F) and 53 (M) were light-responsive, and 29 (F) and 26 (M) were intensity-encoding. The incidence of mPFC persistent light-responsive neurons and of persistent intensity-encoding neurons did not differ between sexes. This information is now stated in the results section (lines 105-106, 143-145) and in Extended Data Figs. 1i and 2e.

Reviewer 2 comment 3:

- In addition to the number of neurons in which light responses were detected, the number of animals in which it was detected should be reported for all analyses (together with the mean \pm SD of neurons per animal). This is not always done and will allow the full appreciation of the generalisation of the findings.

We thank the reviewer for this excellent comment. The revised manuscript now reports the number of mice included in each data set. It also reports the number of mice used for each analysis, and the mean \pm SD incidence of neurons per animal.

Reviewer 2 comment 4:

Likewise, figures should make clear whether the n corresponds to the numbers of neurons or of animals.

Fixed.

Reviewer 2 comment 5:

Likewise supplementary information should detail the number of sessions and neurons per mouse in addition to the number of males and females.

Fixed.

Reviewer 2 comment 6:

- An important limitation of the study is the lack of behavioural measures included. We do not know to which aspect of behaviour the different cell types contribute to and whether they do contribute to any behaviour. This limitation should be clear. Likewise, translation from animal research to humans is not straightforward (e.g. Braak 2015

Brain) and remains to be done such that the authors should be cautious in their statement throughout the text.

We thank the reviewer for this important comment. Being the first report of light encoding neurons in the mPFC of mice, the scope of the current study indeed did not include the further and critical question of what types of behaviours such encoding shapes (a question that we intend to pursue in future studies now that such encoding was revealed). It is yet unknown what, if any, behaviours the observed mPFC photosensitivity contributes to. We have added to the discussion a statement to that effect (lines 484-497). We now further elaborate in the discussion on the limitations of the study in terms of translatability to humans, citing previous works demonstrating these limitations.

Lastly, we scanned the whole text and weakened any potentially too bold a statement regarding the translatability of the findings in mice to humans.

Reviewer 2 comment 7:

- It would be good if figure 6 was to make postulates on which cell type underlies which behaviour. Still on Fig 6, projection f and g are not fully clear (f gets mixed up with l and it's unclear whether g also takes part of f when it crosses it). I recommend using different colours.

We thank the reviewer for this comment. Following the reviewer's comment, selected light-modulated behaviours, and the different mPFC regions and intensity-encoding neuronal types that may underlie these behaviours, have now been added to figure 7 and are described in the figure caption. Additionally, to ensure clarity, projections 'f' and 'g' are now marked in different colours.

Reviewer 2 comment 8:

- Section "Light intensity continuously modulates mPFC neuronal activity" relies on only 3 animals given the small number of mice included.

Following the reviewer's comment, we performed additional recordings in response to the two types of light stimuli (7 intensity steps, bi-phasic stimulus). We doubled the number of mice from 3 to 6, as well as the number of recording sessions from 4 to 13 which resulted in an increase in the number of intensity-encoding neurons from 27 to 55. After accounting for the new data, the previously-proposed conclusions were obtained once more. Accordingly, we have updated the panels of Fig. 3 with the new data set and sample sizes.

Reviewer 2 comment 9:

I wonder whether the lateralisation of some effects could be related to difference in the location of the probes. It should be commented in the text.

We thank the reviewer for this important comment. Following the reviewer's comment, we now report the statistical significance of any differences in the incidence of neurons between hemispheres, across mPFC subregions, across cortical layers, and between

the mPFC and PHb. As explained in the new statistical analysis subsection of the methods section (lines 851-897), when comparing the incidence of light-responsive or intensity-encoding neurons between brain hemispheres or across different brain regions (e.g., mPFC vs. MOs), we tested for statistical significance between the incidences of these neurons per recording session, typically using a permutation t-test or a permutation 1-way ANOVA. This approach, however, was found to be non-optimal for testing for differences in the incidence of light-responsive or intensity-encoding neurons across mPFC subregions or cortical layers. The number of light-responsive or intensity-encoding neurons identified in a given subregion or cortical layer per recording session, and thus also their calculated incidence per session, varied considerably. This high variability resulted from the fact that the multielectrode array, with its 32 electrode sites spanning 750 μm , traversed varying sized-portions of multiple subregions and cortical layers in each recording session. For example, all the recording sessions that targeted the ventral mPFC included only a small portion of the AC, which constitutes the dorsal mPFC. Consequently, in all these sessions, the incidence of light-responsive and intensity-encoding neurons in the AC was low (Extended Data Figure 1l,m). The resulted large variation in the incidence of light-responsive or intensity-encoding neurons per session, led to low a likelihood for detecting statistically-significant differences in the incidence of these neurons between subregions and layers. To reduce our sensitivity to the mentioned variability, when comparing the incidence of light-responsive or intensity-encoding neurons between subregions and layers, we calculated these neurons' incidence as their total number across all recording sessions in a given subregion or layer divided by the total number of neurons identified in that locality across all sessions. Moreover, to determine whether there is a statistically-significant difference between two distributions of incidences, the non-parametric χ^2 (Chi-square) test was utilized.

Because the two approaches account for the number of identified neurons in a given locality, all the presented estimates of neuron incidences throughout the paper are not sensitive to differences in the location of the probe. A similar statement has been added to the methods section (lines 626-630).

Nevertheless, while the incidence of light-responsive and intensity-encoding neurons per recording session did not differ between hemispheres, differences between hemispheres in the incidence of these neurons might still exist between mPFC subregions. To test this, we plotted the incidence of light-responsive and intensity-encoding neurons, per subregion, per hemisphere (Extended Data Figs. 1g and 2g). Indeed, slight differences in lateralization were observed across mPFC subregions.

Reviewer 2 comment 10:

- The author should provide detailed characteristic of the light conditions they used (at least in mel EDI lux) and mention what illuminance they intended to administer and the potential deviations. The precise time of day of the experiments should be provided.

In response to Reviewer 1 comment 1, where similar concerns were indicated, we added a detailed description of the characteristics of the light stimulus. New plots were

added to Extended Data. Fig. 1a,b, and a statement describing the new data has been added to the Methods section. Following the reviewer's comment, to better characterize the light stimuli, we used the melanopic illuminance for rodents toolbox, <https://lucasgroup.lab.manchester.ac.uk/measuringmelanopicilluminance/>) to calculate the melanopic equivalent daylight illuminance (melanopic EDI), which varied from 0.005 to 5506 lux across the 7 presented light intensities. This information has been added to the Methods section (lines 583-586).

To assess the potential deviation in illumination level, we placed a fiber optic at the position of the mouse eye, and performed 10 irradiance measurements following repositioning of the tip of the liquid light guide that delivers the LED light. The standard deviation in melanopsin quantum catches across these 10 measurements ($0.13 \log \text{ photons cm}^{-2} \text{ s}^{-1}$) was taken as the variation in melanopsin quantum catches across recording sessions. This information has been added to the Methods section (lines 588-590).

Recording sessions were performed during the first-half of the dark phase of a 12h/12h light/dark cycle. Recordings commenced at zeitgeber time $16:42 \pm 3:18$ (hour:min, mean \pm s.d.) and lasted 45 ± 3.4 min (mean \pm s.d.). A similar statement has been added to the Methods section (lines 561-563).

Minor

Reviewer 2 comment 11:

- Reference to human studies studying light and mood and/or emotion should be made to strengthen the rationale. Likewise, interpretation would be strengthened by the inclusion of more human studies finding mPFC modulation by light.

Discussion of the effect of light on mPFC-modulated processes and behaviors is included in the first paragraph of the introduction section. Following the reviewer's suggestion, we now expanded this discussion to reference additional previous studies. Moreover, we added to the discussion section a short paragraph on previous reports on the effect of light on the human PFC, and the possible dependency of this effect on ipRGCs drive (lines 469-483).

Reviewer 2 comment 12:

- Environmental light detection relies on slow time scale and this comes at odds with the use of the term "flashes" which should be replaced by "pulses", "blocks" or "exposure"

Following the reviewer's comment, we replaced the term 'flashes' with 'pulses'.

Reviewer 2 comment 13:

- The abstract makes the appealing statement that results could be useful for therapies. However the paper makes no clear statement regarding this aspect.

Following the reviewers' comments, we added a short paragraph in the discussion section (lines 484-497) that points to the intriguing avenue of harnessing the new knowledge of the characteristics of mPFC light responsiveness to develop therapies for mPFC-modulated emotional and cognitive processes and behaviours. We also suggest that the new knowledge on mPFC photosensitivity might be used for the design of optimal light therapy protocols and lighting environments which could potentially be harnessed to alleviating anxiety and addiction disorders, and to the enhancement of cognitive performance.

Reviewer 2 comment 14:

- For some figures, the meaning of * is not provided in the legend

Following the reviewer's comment, we now indicate the definition of asterisks in all figure panels.

Reviewer 2 comment 15:

- Figure display of brain images should specify left and right

Following the reviewer's comment, we added indication of the left and right hemispheres to the brain maps.

Reviewer 2 comment 16:

- The 3rd sentence of the intro is very long and should be broken in smaller pieces for clarity.

Following the reviewer's comment, we split the given statement into two shorter and clearer statements.

Reviewer 2 comment 17:

- Figure 3a, a scale for firing rate should be provided to each panel

Following the reviewer's comment, a scale bar has been added to each of the plots in Fig. 3a.

Reviewer 2 comment 18:

- In what phase of the menstrual cycle were the tested females?

The particular phase of the estrous cycle, which lasts four days in mice, was not documented. However, since each mouse underwent 2-4 recording sessions spanning a week, recordings were likely conducted at multiple phases of the estrous cycle. A similar statement has been added to the 'Animals' subsection of the Methods section (lines 511-514). See also our response to Reviewer 2 Comment 2 for detailed explanation of the limited sex-related data.

Reviewer 2 comment 19:

- Given the masking / sleep-inducing effect of light, the authors should mention mouse behaviour during the recordings at least qualitatively.

We thank the reviewer for this comment. No obvious sleep-inducing effect of light was observed during the recording sessions, possibly due to the uncomfortable posture of the mouse while being head-restrained. Mice appeared to intermittently walk or run on the treadmill, without obvious correlation or anticorrelation to the light stimulus. A similar statement has been added to the *Extracellular in vivo recording in awake mice* section of the Methods section (lines 554-557).

Reviewer #3 (Remarks to the Author):

In this study, the authors investigate light responses and irradiance sensitivity in the mPFC in awake, head-fixed, mice - finding a surprisingly high proportion of cells respond to light across its various subdivisions including a subset that show monotonic changes in firing as a function of irradiance. They also find similar types of responses in the PHb and provide some evidence that a pathway involving ipRGCs and mPFC-projecting PHb neurons might mediate such effects using chemogenetic manipulations and ipRGC ablation.

Overall I find the study interesting. Characterisation of light response types and their anatomical locations is carefully performed and will provide a valuable addition to our understanding of how light influences brain function.

Reviewer 3 comment 1:

This is somewhat let down by the fact that much of the description around anatomical localisation is descriptive without any statistical analysis upon which to assess which differences are meaningful and which aren't. That issue should be easy to address, however.

Following the reviewer's comment, and as mentioned in our response to Reviewer 2 comment 9, we now report the statistical significance of any differences in the incidence of neurons between hemispheres, across mPFC subregions, across cortical layers, and between the mPFC and PHb. As explained in the new statistical analysis subsection of the methods (lines 851-897), when comparing the incidence of light-responsive or intensity-encoding neurons between brain hemispheres or across different brain regions (e.g., mPFC vs. MOs), we performed statistical tests to compare the incidence of these neurons per recording session, typically using a permutation t-test or a permutation 1-way ANOVA. This approach, however, was found to be non-optimal for testing for differences in the incidence of light-responsive or intensity-encoding neurons across mPFC subregions or cortical layers. The number of light-responsive or intensity-encoding neurons identified in a given subregion or cortical layer per recording session, and thus also their calculated incidence per session, varied considerably. This high variability resulted from the fact that the multielectrode array, with its 32 electrode sites spanning 750 μm , traversed different portions of multiple subregions and cortical layers in each recording session. For example, all the recording sessions that targeted the ventral mPFC included only a small portion of the AC, which constitutes the dorsal mPFC. In all these sessions, the incidence of light-responsive and intensity-encoding neurons in the AC was low (Extended Data Figure 1l,m). However, our ability to conclude from this that the incidence of light-encoding neurons in the AC is lower than in other areas is very limited, since it is more likely that the low incidence resulted from smaller coverage of the AC by the electrode sites. The resulting large variation in the incidence of light-responsive or intensity-encoding neurons per session, led to low likelihood for detecting statistically-significant differences in the incidence of these neurons between subregions and layers. To reduce our sensitivity to the mentioned variability, when comparing the incidence of light-responsive or intensity-encoding

neurons between subregions and layers, we calculated these neurons' incidence as their total number across all recording sessions in a given subregion or layer divided by the total number of neurons identified in that locality across all sessions. Moreover, to determine whether there is a statistically-significant difference between two distributions of incidences, the non-parametric χ^2 (Chi-square) test was utilized.

Reviewer 3 comment 2:

A more substantial and harder to address issue relates to the circuit mapping aspects of the study:

1) Given that these are awake/behaving animals can the authors exclude the possibility that light responses observed are secondary to some general behavioural or physiological responses to light rather than specific pathways carrying light information (eg ipRGC->PHb->mPFC)

The reviewer raises an important issue that has not been sufficiently addressed in the original manuscript. The responses of the mPFC to sensory stimuli are, by definition, secondary in nature, as the mPFC does not receive any direct sensory input. Instead, the mPFC integrates input from a variety of brain regions, with the activity in some of them being modulated by light exposure, e.g., the central amygdala and lateral hypothalamus that are innervated by ipRGCs, as indicated in Fig. 7. Other brain regions that derive light sensitivity from conventional RGCs might also contribute to mPFC photosensitivity. Additionally, light alters a multitude of physiological processes and behaviours, which in turn may modulate mPFC activity. For example, light may increase fear by modulating activity in the mPFC-projecting basolateral amygdala, which in turn might affect mPFC activity. Importantly, regardless of whether the effect of light is primary or of any other order, our results demonstrate the ability of light to modulate the activity of selected mPFC neurons, and the ability of these neurons to persistently encode the intensity of light. A similar statement has been added to the discussion section (lines 418-424).

Reviewer 3 comment 3:

2) For chemogenetic studies such as those employed here it is not sufficient to rely on comparisons between CNO and saline. Given that it is now widely accepted that CNO can exert off-target (non-DREADD-mediated) effects, really these studies need a non-DREADD vector control group (eg mCherry) that also receives CNO. Note also that following point 1, above, the DREADD manipulations if proved effective could still be acting via indirect behavioural and/or physiological effects (eg changes in running speed, pupil size etc.) which would need to be controlled for.

We thank the reviewer for this important observation. Following the reviewer's comment, as a control for the ipRGCs-inhibiting experiment, we performed a set of control experiments in which mCherry, instead of DREADD, was virally expressed in ipRGCs. We then recorded light-evoked neuronal firing rate in the mPFC after saline or CNO infusion. Specifically, among light-responsive neurons, we compared the firing rate change in response to the highest light intensity in DREADD-expressing vs. mCherry-

expressing neurons, following infusion of saline vs. CNO. This revealed a significant effect of CNO vs. saline in DREADD-expressing mice, but not in mCherry-expressing mice. Among intensity-encoding neurons, we compared the firing rate in response to the 7 tested light intensities, in DREADD-expressing vs. mCherry-expressing mice, after saline vs. CNO infusion. In DREADD-expressing mice, firing rate following saline infusion, but less so following CNO infusion, increased monotonically with light intensity. In mCherry-expressing mice, firing rate increased with light intensity following either saline or CNO infusion. This suggests that the observed decrease in intensity-encoding following CNO is mainly due to chemogenetically inhibiting the DREADD-expressing ipRGCs. The new results are now described in the results text (lines 310-342) and are presented in Fig. 6.

The study includes another chemogenetic manipulation experiment for inhibiting the PHb. To control for effects that CNO may exert on its own on mPFC photosensitivity, we recorded mPFC light-evoked activity following CNO or saline infusion to mice that do not express DREADD in the PHb. The incidence of persistent light-responsive mPFC neurons following CNO infusion was significantly lower than following saline infusion in the DREADDs group. However, the incidence of such neurons did not differ significantly between saline and CNO infusion in the control group. Thus, the light-responsiveness of the mPFC derives, at least in part, either directly or indirectly, from the PHb. The new results are now described in the results text (lines 228-249) and are presented in Fig. 4d.

The effect of CNO-induced inhibition of ipRGCs on locomotion was not assessed. However, no obvious changes in locomotion were observed following saline or CNO infusion. A similar statement has been added to the Methods section (lines 646-647).

CNO-induced inhibition of ipRGCs dilates the pupil, as previously shown by Storchi et al. (2015) (<https://pubmed.ncbi.nlm.nih.gov/26438865/>). Pupil dilation would allow a larger amount of light to enter the eye, potentially leading to greater activation of ipRGCs and the light-sensitive mPFC neurons they drive. Indeed, as we now show, pharmacologically dilating the pupils slightly increased the FR of light-responsive and intensity-encoding mPFC neurons, albeit not at a statistically-significant level (Extended Data Fig. 9d). In contrast, CNO infusion in our chemogenetic experiments, that also dilated the pupil, decreased the firing rate of mPFC neurons, suggesting that the observed effect of CNO infusion on mPFC firing cannot be explained by an effect of CNO-induced inhibition of ipRGCs on pupil size. A similar statement has been added to the Methods section (lines 647-656).

Reviewer 3 comment 4:

Leaving those points aside, while I agree the data are potentially consistent with some effect of inhibiting/lesioning ipRGCs (and to a lesser extent also inhibiting mPFC-projecting PHb cells), the effects are not especially compelling. This may be due to low n numbers and/or low penetrance of the manipulations. Either way, I think it would be hard to be certain from the present data that ipRGC->PHb is the primary circuit that mediates effects of light/irradiance on the mPFC. Indeed it is notable that effects of light

on the mPFC are substantially slower than those in the PHb (40ms as reported here). That seems like quite a long delay for a monosynaptic connection. The study would be substantially enhanced, if data specifically linking the PHb projection to light induced changes in the mPFC could be strengthened and/or data on the physiological relevance of mPFC light responses could be provided.

Following the reviewer's insightful comment, we performed a large number of additional experiments to test the dependency of mPFC photosensitivity on ipRGCs and PHb drive.

Two different approaches to test the contribution of ipRGCs to mPFC photosensitivity

1. To test the effect of dtA-induced terminal ipRGCs ablation on mPFC photosensitivity, we performed additional experiments that increased the number of neurons 3-fold. This expanded data set, compared to the original data set, includes 6 vs. 3 mice, 15 vs. 5 recording sessions, 473 vs. 171 neurons, 92 vs. 38 persistent light-responsive neurons, and 26 vs. 8 persistent intensity-encoding neurons. These new results are now described in the results section (lines 343-370) and presented in Fig. 6g-k.
2. For the same study of dtA-induced terminal ablation of ipRGCs, we added a control experiment to test for an effect of the viral eye-injections on mPFC photosensitivity. We bilaterally injected *Opn4^{Cre/+}* mice with an AAV inducing Cre-dependent expression of mCherry. This 'mCherry' group includes 158 neurons identified in 6 recording sessions and 3 mice. Therefore, in the revised manuscript, the dtA-induced terminal ablation data have been compared to two control groups: the 'mCherry' group and a naïve wild type mice group (main data set). We found that terminal ablation of ipRGCs significantly decreased mPFC photosensitivity.
3. As specified in our response to Reviewer 3 comment 3, for the ipRGCs chemogenetic inhibition experiment, we performed a new set of experiments to control for any DREADD-independent effects of CNO. In these control experiments, mCherry, instead of DREADD, was virally expressed in ipRGCs (3 mice, 6 recording sessions, 56 light-responsive neurons). We then recorded light-evoked neuronal firing rate in the mPFC after saline or CNO infusion. Comparing the DREADD and mCherry groups demonstrated that the observed decrease in intensity-encoding following CNO is mainly due to chemogenetically inhibiting the DREADD-expressing ipRGCs. These new results are now described in the results section (lines 310-342) and presented in Fig. 6a-f.

Two different approaches to test the contribution of the PHb to mPFC photosensitivity

4. As specified in our response to Reviewer 3 comment 3, for the experiment of chemogenetic inhibition of mPFC-projecting PHb, to control for any effect CNO might exert on its own on mPFC photosensitivity, we recorded mPFC light-evoked activity following CNO or saline infusion to mice that do not express DREADD in the PHb. The incidence of persistent light-responsive mPFC neurons following CNO infusion was significantly lower than following saline infusion in the DREADDs group, but not

in the control group. These new results are now described in the results section (lines 228-249) and presented in Fig. 4a-d.

5. In addition to the original experiments of chemogenetic inhibition of mPFC-projecting PHb neurons, we added an experiment of optogenetic inhibition of PHb neurons. We virally expressed in the PHb the inhibiting light-activated G protein-coupled receptor, the eOPN3. Then, by inserting an optotrode into the mPFC, we photoactivated the axon terminals of projecting PHb neurons to achieve continuous inhibition of the axon terminals, and simultaneously, recorded light-evoked firing in mPFC neurons. Since both eOPN3 and the pathways underlying mPFC photosensitivity are modulated by light, we did not attempt comparing light-evoked mPFC firing rate before vs. during eOPN3 optogenetic activation. Instead, we recorded light-evoked mPFC firing under sustained eOPN3-induced inhibition of axon terminals. The eOPN3 group included 221 neurons identified in 5 recording sessions and 3 mice. These new results are now described in the results section (lines 250-276) and presented in Fig. 4e-i.
6. To control for an eOPN3-independent effect of the photoactivation, we used the same optotrode and the same photoactivation/recording protocol in mice in which the fluorescent reporter mScarlet, instead of eOPN3, was virally expressed in PHb neurons. The mScarlet group included 145 neurons identified in 4 recording sessions and 2 mice. We found that (1) steady-state firing rate in the eOPN3 group was lower than in the mScarlet group; (2) the percentage of persistent light-responsive and intensity-encoding neurons in the eOPN3 group was significantly lower than in the mScarlet group, and (3) the percentage of each of the four mPFC intensity-encoding neuronal types, was lower in the eOPN3 than in the mScarlet group.

Together, the original and new experiments demonstrate, we believe, that ipRGCs and the PHb contribute to the light-responsiveness of the mPFC. This however does not suggest that mPFC photosensitivity derives exclusively from ipRGCs and the PHb, nor that mPFC photosensitivity depends on transmission of ipRGCs light signals through the PHb. Instead, mPFC photosensitivity is likely also shaped by light signals arriving from other brain areas or other RGC types. Fig. 7 indicates the possible involvement of several such areas and RGCs in shaping mPFC photosensitivity. A similar statement has been added to the discussion section (lines 377-384).

Additionally, we modified any statement, throughout the manuscript, that might be misunderstood as though we argue that mPFC photosensitivity depends solely on ipRGCs signaling through the PHb.

Specific comments:

Reviewer 3 comment 6:

P4: 'The incidence of transient and persistent light-responsive neurons was highest in the AC, PL, and IL (transient: 43-49%, persistent: 34-42%, of identified neurons) (Fig. 1f)' – do authors find evidence that incidence varies significantly as a function of subregion? In the absence of any statistical test to exclude the alternate possibility that

this just reflects a random sampling artefact this sentence should be removed or revisited.

Following the reviewer's comment, we now report the statistical significance of any differences in the incidence of neurons between hemispheres, across mPFC subregions, across cortical layers, and between the mPFC and PHb. This is now explained in detail in our response to Reviewer 3 comment 1, and in the new statistical analysis subsection of the methods section (lines 853-899).

Reviewer 3 comment 7:

P4: 'The incidence of light responsive neurons also varied across hemispheres, being larger in the left hemisphere (46% transient and 39% persistent in the left hemisphere, vs. 38% transient and 32% persistent in the right hemisphere) (Fig. 1g)' –as above, is this actually a meaningful difference?

As detailed in the new statistical analysis subsection of the methods section, when comparing the incidence of light-responsive or intensity-encoding neurons between brain hemispheres, we tested for statistical significance between the incidences of these neurons per recording session, using a permutation t-test. According to this test, and in contrast to our original report, the incidence of light-responsive neurons and persistent intensity-encoding neurons did not differ significantly across hemispheres. The new results are presented in the text (lines 101-104, 143-146) and in Figs. 1i and 2k.

Reviewer 3 comment 8:

P4: 'An important characteristic of a system capable of mediating an effect of light on emotion and cognition would be the ability to persistently encode the intensity of light.' – I This is a bit circular, effects of 'light' on emotion and cognition needn't necessarily be restricted to modulation according to ambient light intensity.

We agree with the reviewer's suggestion that effects of light on emotion and cognition need not necessarily be restricted to modulation according to ambient light intensity. Indeed, light vs. darkness might theoretically exert a binary effect on physiology and behaviour. However, encoding light intensity would allow also gradual modulation of physiology and behaviour according to light intensity. To clarify this important point, we added similar statements to the results section (lines 109-114).

Reviewer 3 comment 9:

P5: 'Persistent intensity-encoding neurons were found throughout the mPFC (Fig. 2i), with a slightly higher incidence in the left hemisphere (17%, vs. 12% in the right hemisphere). The regions with the highest incidence of these neurons were the PL and the IL' – Same comments as above apply to much of this whole paragraph. A lot of qualitative statements are made which might simply reflect random sampling effects rather than meaningful differences, especially given the low numbers of cells in each class.

As indicated in our response to Reviewer 2 comment 3 and Reviewer 3 comment 1, we now report the statistical significance of any differences in the incidence of neurons between hemispheres, across mPFC subregions, across cortical layers, and between the mPFC and PHb. The selection of statistical tests is thoroughly explained in the new statistical analysis subsection of the methods section (lines 853-899).

Reviewer 3 comment 10:

P5: 'Analysing the (darkness) spontaneous firing and action potential waveforms revealed that the four types corresponded to mixtures of presumptive pyramidal neurons and interneurons' – I think this statement is a bit strong. I find the analysis in extended figure 2 convincing that this is more than one group (perhaps 3 based on panel d?) and one would assume the main group is pyramidal neurons. Nonetheless, given it is hard to be certain about this classification, I think suggested would be better than 'revealed'

Following the reviewer's comment, we replaced the term 'revealed' with 'suggested'. Indeed, while the main cluster most likely corresponds to pyramidal neurons, the remaining neurons might belong to different types of inhibitory interneurons that aren't easily separable based on waveform statistics. A similar statement has been added to the results section (lines 163-174).

Reviewer 3 comment 11:

P6, 1st paragraph – again many qualitative statements that are hard to evaluate without any statistical analysis.

We thank the reviewer for this comment. We now perform statistics on the two main claims presented in the mentioned paragraph. First, to evaluate each type's capacity to track intensity gradients, we fitted a sigmoid to the firing rate encountered throughout each phase, for each type. The fit was statistically significant for all four types, demonstrating they all continuously encode light intensity. Second, to test the significance of the effect of light history on the firing rate of intensity-encoding mPFC neurons, we compared the firing rate over the 7 sec after the transition time (when stimulus intensity exceeds $15 \log \text{ photons cm}^{-2} \text{ s}^{-1}$) and that over the 7 sec prior the transition time. The firing rate after the transition time was significantly higher than that prior the transition time. This was true when (1) probing all neurons of all types pooled, (2) probing the 'enhanced' and 'suppressed' types separately, and (3) probing the 'ON-OFF' and 'ON' types separately. These results demonstrate a significant effect of light history on neuronal firing rate. The new data are described in the results section (lines 195-208) and in Fig. 3 and Supplementary Fig. 4).

Reviewer 3 comment 12:

P7- 'Dependence of mPFC photosensitivity on input from the presynaptic PHb' section – I find this section very hard to follow. Fig 4a starts with the experimental paradigm for viral labelling and example image of transduced neurons. The rest of the data shown in

this figure, so far as I can gather, compares the properties of PHb and mPFC cells but has nothing to do with the DREADD manipulation per se.

We completely agree with the reviewer's comment. We have now moved the experimental paradigm for viral labelling and example images of transduced neurons to Extended Data Fig. 6 that actually displays the data related to PHb chemogenetic inhibition. Additionally, we have now added to Fig. 4 a new schematic showing the experimental setup for extracellular recordings from the PHb.

Reviewer 3 comment 13:

There is some data on effects of the DREADD manipulation of PHb on mPFC in extended data fig 7 but that is hard to evaluate and is not especially convincing as currently presented, nor is the paradigm explained in sufficient detail. Are the same cells tested both under saline and CNO? If so is the order of testing randomised? Did the authors validate whether the approach actually silenced PHb neurons?

As described in our response to Reviewer 3 comment 4, we performed additional experiments, and the revised manuscript now reports two different approaches to test the contribution of the PHb to mPFC photosensitivity. For the experiment of chemogenetic inhibition of mPFC-projecting PHb, to control for any effect CNO might exert on its own on mPFC photosensitivity, we recorded mPFC light-evoked activity following CNO or saline infusion to mice that do not express DREADD in the PHb. The incidence of persistent light-responsive mPFC neurons following CNO infusion was significantly lower than following saline infusion in the DREADDs group, but not in the control group. We also added an experiment of optogenetic inhibition of PHb neurons to further test the contribution of PHb drive to mPFC photosensitivity. We virally expressed in the PHb the inhibiting light-activated eOPN3. Then, by inserting an optotrode into the mPFC, we photoactivated the axon terminals of projecting PHb neurons to achieve continuous inhibition of the axon terminals, and simultaneously, recorded light-evoked firing in mPFC neurons. To control for an eOPN3-independent effect of the photoactivation, we used the same optotrode and the same photoactivation/recording protocol in mice in which the fluorescent reporter mScarlet was virally expressed in PHb neurons, instead of eOPN3. As specified in the results and in our response to Reviewer 3 comment 4, these optogenetic inhibition demonstrated that mPFC photosensitivity derives, at least in part from the PHb, whether directly or indirectly.

In our chemogenetic inhibition experiments, the light-evoked responses of the same mPFC neurons were tested both under saline and CNO. To ensure clarity, we added a detailed description of the paradigm to the caption of Fig. 4. Firing rate in response to 7 light intensities was recorded twice, first 10 min following saline infusion, and again 10 min following CNO infusion. Reversing the order of saline and CNO infusion was not possible because the CNO effect might linger for more than an hour, much longer than the possible duration of an acute recording session. A similar statement, with references, has been added to the methods section (lines 633-645).

Validating the CNO-induced inhibition of PHb neurons is challenging if not impossible. A reduction in the firing rate of PHb neurons might point to an effect of CNO, but might also be a result of minute movements of the multi electrode array, which often lead to the “loss of the cells”. Since there is no proven way to distinguish between these two situations, validating the inhibition of PHb neurons was not attempted. A similar statement has been added to the methods section (lines 677-680). Utilizing *in vitro* whole-cell recordings from PHb neurons in brain slices following superfusion of CNO vs. aCSF may also be applied. However, the continuous superfusion of CNO into the bath would differ considerably from the acute subcutaneous infusion of CNO in the *in vivo* experiments. Therefore, this approach would not validate the inhibition of PHb by CNO in the intact mouse. The same limitation is found in *in vitro* whole-cell recordings in isolated retinas, which we applied to test the effect of CNO on DREADD-expressing ipRGCs (Extended Data Fig. 8). Those experiments showed that continuous CNO superfusion indeed inhibits ipRGC activity. However, also in this case, it is not entirely clear how the results of such an *in vitro* assay would predict the effect of acute CNO infusion in the intact mouse. Importantly, acknowledging that the chemogenetic inhibition experiments originally presented were weaker than desired, we decided to utilize a completely different approach to further test the dependence of mPFC photosensitivity on PHb input.

Reviewer 3 comment 14:

In terms of the data presented, there is information on % cells under saline vs. CNO but actual cell numbers are unclear, and there is no statistical analysis reported. Given the low n numbers for subsequent panels (e.g; n=9 and n=3 respectively), I suspect the percentages shown in panels a-d must come from a very low overall number of cells and so I wonder whether statistical tests would reveal any meaningful difference. The average response profiles shown in panels e and f suggest maybe a modest reduction for suppressed cells (e), while enhanced responses (f) are so weak under baseline conditions it is hard to say whether there is any meaningful effect at all.

Following the reviewer’s comment, and considering the small number of neurons included in the analysis of the effect of CNO of the spontaneous and light-evoked firing of mPFC neurons, we decided to omit the two plots describing these results. Instead, we focused on the effect of CNO vs. saline on the firing rate of all light-responsive neurons (now Fig. 6c).

Reviewer 3 comment 15:

P8: ‘Analysing the spontaneous firing activity and action potential waveforms of identified PHb neurons again yielded two distinct neuronal populations’ – the analysis shown does not, to me, convince that there are two distinct populations (as opposed to a single population with a long tail), nor do I think it valid to assume that cells below an arbitrary cutoff are inhibitory. As a more general point, the use of spike shape to identify inhibitory interneurons may have validity for cortex and has (as the authors note) also been used for other brain regions. I don’t believe the reliability of this approach has ever actually been validated for subcortical regions however. As such the section of the manuscript and associated data figure is highly speculative.

We agree with the reviewer on the weakness of the approach presented in this Supplementary Figure. Therefore, we have decided to omit this figure all together. Fortunately, considering that excitatory neurons represent ~74% of PHb neurons, and that extracellular recordings tend to detect more neurons than interneurons, as neurons typically have larger and more easily detectable spikes than interneurons, excitatory neurons are likely to dominate the neuronal population we detected in the PHb. As a consequence, omitting the figure does not affect our interpretation of the results. A similar statement has now been added to the discussion section (lines 399-409).

Reviewer 3 comment 16:

Fig 2: axes on k-m are misleading. If I understand what is being shown correctly, for k it would be appropriate to use % identified neurons (as in Figure 1). For l-m it should be % of irradiance coding neurons (or similar). Indeed for the latter that actual percentages of all identified cells are very small – in that regard it might be more informative to show the raw proportions.

Following the reviewer's comment, the axes titles have been corrected in Figs. 2, 5, and 6, and in Extended Data Figs. 2 and 5. Additionally, we calculated the incidences of the 'enhanced' and 'suppressed' neurons out of all identified neurons (Supplementary Fig. 2h), in line with the reviewer's suggestion. With this new calculation, as before, the IL and DP were dominated by the two enhancement-response types, and the PL and dTT by the two suppression-response types.

Reviewers' Comments:

Reviewer #1:

Remarks to the Author:

The authors answered all my questions satisfactorily. I congratulate the authors on this very interesting study which sheds light on the mechanisms of non-image-forming effects of light in the prefrontal cortex. I hope that there will be some translational aspects from these results that may be transferred to the treatment of depression and anxiety disorders in humans.

Reviewer #2:

Remarks to the Author:

After careful consideration of the responses to the issues I had raised as well as to the responses to the comments of the other 2 reviewers, I have no further comments. The authors satisfactorily addressed my comments and I congratulate them for the additional work completed lead to what I think is a very interesting piece of work.

Reviewer #3:

Remarks to the Author:

The Authors have been very responsive to comments raised at the previous review stage, including the addition of useful additional data and analysis. It would have been nice if the added optogenetic studies included before vs. after comparisons (which would be a more powerful demonstration than the between animal studies). As it stands I think the data support the view that peri-habenlula contributes to mPFC light responses but leave the extent to which they are important (vs. other potential routes) unclear. Nonetheless, I feel this level of uncertainty is appropriately dealt with in the manuscript.

As such I have no further major points to address. I would, however, encourage the authors to ensure it is clear how many mice and recordings sessions contribute to the various experiments. This information is provided in many but not all places (so far as I can see) e.g. I couldn't find numbers of mice for the optogenetic experiments.

REVIEWERS' COMMENTS

Reviewer #1:

The authors answered all my questions satisfactorily. I congratulate the authors on this very interesting study which sheds light on the mechanisms of non-image-forming effects of light in the prefrontal cortex. I hope that there will be some translational aspects from these results that may be transferred to the treatment of depression and anxiety disorders in humans.

Reviewer #2:

After careful consideration of the responses to the issues I had raised as well as to the responses to the comments of the other 2 reviewers, I have no further comments. The authors satisfactorily addressed my comments and I congratulate them for the additional work completed lead to what I think is a very interesting piece of work.

Reviewer #3:

The Authors have been very responsive to comments raised at the previous review stage, including the addition of useful additional data and analysis. It would have been nice if the added optogenetic studies included before vs. after comparisons (which would be a more powerful demonstration than the between animal studies). As it stands I think the data support the view that peri-habenlula contributes to mPFC light responses but leave the extent to which they are important (vs. other potential routes) unclear. Nonetheless, I feel this level of uncertainty is appropriately dealt with in the manuscript.

As such I have no further major points to address. I would, however, encourage the authors to ensure it is clear how many mice and recordings sessions contribute to the various experiments. This information is provided in many but not all places (so far as I can see) e.g. I couldn't find numbers of mice for the optogenetic experiments.

Response:

The numbers of mice, recording sessions and neurons for the optogenetic experiments are specified in the caption of Fig. 4g.